# AQUILA: Communication Efficient Federated Learning with Adaptive Quantization of Lazily-Aggregated Gradients

## Abstract

The development and deployment of federated learning (FL) have been bottle-necked by heavy communication overheads of high-dimensional models between the distributed device nodes and the central server. To achieve better error-communication trade-offs, recent efforts have been made to either adaptively reduce the communication frequency by skipping unimportant updates, e.g., lazy aggregation, or adjust the quantization bits for each communication. In this paper, we propose a unifying communication efficient framework for FL based on **a**daptive **qu**antization of **l**azily-**a**ggregated gradients (**AQUILA**), which adaptively balances two mutually-dependent factors, the communication frequency, and the quantization level. Specifically, we start with a careful investigation of the classical lazy aggregation scheme and formulate AQUILA as an optimization problem where the optimal quantization level is selected by minimizing the model deviation caused by update skipping. Furthermore, we devise a new lazy aggregation strategy to better fit the novel quantization criterion and retain the communication frequency at an appropriate level. The effectiveness and convergence of the proposed AQUILA framework are theoretically verified. The experimental results demonstrate that AQUILA can reduce around **60%** of overall transmitted bits compared to existing methods while achieving identical model performance in a number of non-homogeneous FL scenarios, including Non-IID data and heterogeneous model architecture.

## 1 Introduction

With the deployment of ubiquitous sensing and computing devices, the Internet of things (IoT), as well as many other distributed systems, have gradually grown from concept to reality, bringing dramatic convenience to people's daily life (Du et al., 2020; Liu et al., 2020; Hard et al., 2018). To fully utilize such distributed computing resources, distributed learning provides a promising framework that can achieve comparable performance with the traditional centralized learning scheme. However, the privacy and security of sensitive data during the updating and transmission processes in distributed learning have been a growing concern. In this context, *federated learning (FL)* (McMahan et al., 2017) has been developed, allowing distributed devices to collaboratively learn a global model without privacy leakage by keeping private data isolated and masking transmitted information with secure approaches. On account of its privacy-preserving property and great potentiality in some distributed but privacy-sensitive fields such as finance and health, FL has attracted tremendous attention from both academia and industry in recent years.

Unfortunately, in many FL applications, such as image classification and objective recognition, the trained model tends to be high-dimensional, resulting in significant communication costs. Hence, communication efficiency has become one of the key bottlenecks of FL. To this end, Sun et al. (2020) proposes the lazily-aggregated quantization (LAQ) method to skip unnecessary parameter uploads by estimating the value of gradient innovation — the difference between the current unquantized gradient and the previously quantized gradient. Moreover, Mao et al. (2021) devises an adaptive quantized gradient (AQG) strategy based on LAQ to dynamically select the quantization level within some artificially given numbers during the training process. Nevertheless, the AQG is still not sufficiently adaptive because the pre-determined quantization levels are difficult to choose in complicated FL

environments. In another separate line of work, Jhunjhunwala et al. (2021) introduces an adaptive quantization rule for FL (AdaQuantFL), which searches in a given range for an optimal quantization level and achieves a better error-communication trade-off.

Most previous research has investigated optimizing communication frequency or adjusting the quantization level in a highly adaptive manner, but not both. Intuitively, we ask a question, *can we adaptively adjust the quantization level in the lazy aggregation fashion to simultaneously reduce transmitted amounts and communication frequency?* In the paper, we intend to select the **optimal quantization level** for every participated device by optimizing the model deviation caused by skipping quantized gradient updates (i.e., lazy aggregation), which gives us a novel quantization criterion cooperated with a new proposed **lazy aggregation strategy** to reduce overall communication costs further while still offering a convergence guarantee. The contributions of this paper are trifold.

- We propose an innovative FL procedure with **a**daptive **qu**ant**i**zation of **l**azily-**a**ggregated gradients termed AQUILA, which simultaneously adjusts the communication frequency and quantization level in a synergistic fashion.

- Instead of naively combining LAQ and AdaQuantFL, AQUILA owns a completely different device selection method and quantitative level calculation method. Specifically, we derive an adaptive quantization strategy from a new perspective that minimizes the model deviation introduced by lazy aggregation. Subsequently, we present a new lazy aggregation criterion that is more precise and saves more device storage. Furthermore, we provide a convergence analysis of AQUILA under the generally non-convex case and the Polyak-Łojasiewicz condition.

- Except for normal FL settings, such as independent and identically distributed (IID) data environment, we experimentally evaluate the performance of AQUILA in a number of **non-homogeneous** FL settings, such as non-independent and non-identically distributed (Non-IID) local dataset and various heterogeneous model aggregations. The evaluation results reveal that AQUILA considerably mitigates the communication overhead compared to a variety of state-of-art algorithms.

## 2 BACKGROUND AND RELATED WORK

Consider an FL system with one central parameter server and a device set $\mathcal{M}$ with $M = |\mathcal{M}|$ distributed devices to collaboratively train a global model parameterized by $\boldsymbol{\theta} \in \mathbb{R}^d$. Each device $m \in \mathcal{M}$ has a private local dataset $\mathcal{D}_m = \{(\boldsymbol{x}_1^{(m)}, \boldsymbol{y}_1^{(m)}), \cdots, (\boldsymbol{x}_{n_m}^{(m)}, \boldsymbol{y}_{n_m}^{(m)})\}$ of $n_m$ samples. The federated training process is typically performed by solving the following optimization problem

$$\min_{\boldsymbol{\theta} \in \mathbb{R}^d} f(\boldsymbol{\theta}) = \frac{1}{M} \sum_{m=1}^{M} f_m(\boldsymbol{\theta}) \quad \text{with} \quad f_m(\boldsymbol{\theta}) = [l(h_{\boldsymbol{\theta}}(\boldsymbol{x}), \boldsymbol{y})]_{(\boldsymbol{x}, \boldsymbol{y}) \sim \mathcal{D}_m}, \tag{1}$$

where $f : \mathbb{R}^d \to \mathbb{R}$ denotes the empirical risk, $f_m : \mathbb{R}^d \to \mathbb{R}$ denotes the local objective based on the private data $\mathcal{D}_m$ of the device $m$, $l$ denotes the local loss function, and $h_{\boldsymbol{\theta}}$ denotes the local model. The FL training process is conducted by iteratively performing local updates and global aggregation as proposed in (McMahan et al., 2017). First, at communication round $k$, each device $m$ receives the global model $\boldsymbol{\theta}^k$ from the parameter server and trains it with its local data $D_m$. Subsequently, it sends the local gradient $\nabla f_m(\boldsymbol{\theta}^k)$ to the central server, and the server will update the global model with learning rate $\alpha$ by

$$\boldsymbol{\theta}^{k+1} := \boldsymbol{\theta}^k - \frac{\alpha}{M} \sum_{m \in \mathcal{M}} \nabla f_m(\boldsymbol{\theta}^k). \tag{2}$$

**Definition 2.1 (Quantized gradient innovation).** *For more efficiency, each device only uploads the quantized deflection between the full gradient $\nabla f_m(\boldsymbol{\theta}^k)$ and the last quantization value $\boldsymbol{q}_m^{k-1}$ utilizing a quantization operator $\mathcal{Q} : \mathbb{R}^d \to \mathbb{R}^d$, i.e.,*

$$\Delta \boldsymbol{q}_m^k = \mathcal{Q}(\nabla f_m(\boldsymbol{\theta}^k) - \boldsymbol{q}_m^{k-1}). \tag{3}$$

For communication frequency reduction, the **lazy aggregation** strategy allows the device $m \in \mathcal{M}$ to upload its newly-quantized gradient innovation at epoch $k$ only when the change in local gradient is

sufficiently larger than a threshold. Hence, the quantization of the local gradient $\boldsymbol{q}_m^k$ of device $m$ at epoch $k$ can be calculated by

$$\boldsymbol{q}_m^k := \begin{cases} \boldsymbol{q}_m^{k-1}, & \text{if } \left\| \mathcal{Q}(\nabla f_m\left(\boldsymbol{\theta}^k\right) - \boldsymbol{q}_m^{k-1}) \right\|_2^2 \leqslant \text{Threshold} \\ \boldsymbol{q}_m^{k-1} + \Delta \boldsymbol{q}_m^k, & \text{otherwise} \end{cases}. \qquad (4)$$

If the device $m$ skips the upload of $\Delta \boldsymbol{q}_m^k$, the central server will reuse the last gradient $\boldsymbol{q}_m^{k-1}$ for aggregation. Therefore, the global aggregation rule can be changed from (2) to:

$$\boldsymbol{\theta}^{k+1} = \boldsymbol{\theta}^k - \frac{\alpha}{M} \sum_{m \in \mathcal{M}} \boldsymbol{q}_m^k = \boldsymbol{\theta}^k - \frac{\alpha}{M} \sum_{m \in \mathcal{M}^k} \left( \boldsymbol{q}_m^{k-1} + \Delta \boldsymbol{q}_m^k \right) - \frac{\alpha}{M} \sum_{m \in \mathcal{M}_c^k} \boldsymbol{q}_m^{k-1}, \qquad (5)$$

where $\mathcal{M}^k$ denotes the subset of devices that upload their quantized gradient innovation, and $\mathcal{M}_c^k = \mathcal{M} \setminus \mathcal{M}^k$ denotes the subset of devices that skip the gradient update and reuse the old quantized gradient at epoch $k$.

For AdaQuantFL, it is proposed to achieve a better error-communication trade-off by adaptively adjusting the quantization levels during the FL training process. Specifically, AdaQuantFL computes the optimal quantization level $(b^k)^*$ by $(b^k)^* = \lfloor \sqrt{f(\boldsymbol{\theta}^0)/f(\boldsymbol{\theta}^k)} \cdot b_0 \rfloor$, where $f(\boldsymbol{\theta}^0)$ and $f(\boldsymbol{\theta}^k)$ are the global objective loss defined in (1).

However, AdaQuantFL transmits quantized gradients at **every communication round**. In order to skip unnecessary communication rounds and adaptively adjust the quantization level for each communication jointly, a naive approach is to quantize lazily aggregated gradients with AdaQuantFL. Nevertheless, it fails to achieve efficient communication for several reasons. First, given the descending trend of training loss, AdaQuantFL's criterion may lead to a high quantization bit number even exceeding 32 bits in the training process (assuming a floating point is represented by 32 bits in our case), which is too large for cases where the global convergence is already approaching and makes the quantization meaningless. Second, a higher quantization level results in a smaller quantization error, leading to a lower communication threshold in the lazy aggregation criterion (4) and thus a higher transmission frequency.

Consequently, it is desirable to develop a more efficient adaptive quantization method in the lazily-aggregated setting to improve communication efficiency in FL systematically.

## 3 Adaptive Quantization of Lazily-Aggregated Gradients

Given the above limitations of the naive joint use of the existing adaptive quantization criterion and lazy aggregation strategy, this paper aims to design a unifying procedure for communication efficiency optimization where the quantization level and communication frequency are considered synergistically and interactively.

### 3.1 Optimal quantization level

First, we introduce the definition of a deterministic rounding quantizer and a fully-aggregated model.

**Definition 3.1. (Deterministic mid-tread quantizer.)** *Every element of the gradient innovation of device $m$ at epoch $k$ is mapped to an integer $[\boldsymbol{\psi}_m^k]_i$ as*

$$\left[ \boldsymbol{\psi}_m^k \right]_i = \left\lfloor \frac{\left[ \nabla f_m(\boldsymbol{\theta}^k) \right]_i - \left[ \boldsymbol{q}_m^{k-1} \right]_i + R_m^k}{2\tau_m^k R_m^k} + \frac{1}{2} \right\rfloor, \forall i \in \{1, 2, ..., d\}, \qquad (6)$$

where $\nabla f(\boldsymbol{\theta}_m^k)$ denotes the current unquantized gradient, $R_m^k = \|\nabla f_m(\boldsymbol{\theta}^k) - \boldsymbol{q}_m^{k-1}\|_\infty$ denotes the quantization range, $b_m^k$ denotes the quantization level, and $\tau_m^k := 1/(2^{b_m^k} - 1)$ denotes the quantization granularity. More explanations on this quantizer are exhibited on Appendix A.2.

**Definition 3.2 (Fully-aggregated model).** *The fully-aggregated model $\tilde{\boldsymbol{\theta}}$ without lazy aggregation at epoch $k$ is computed by*

$$\tilde{\boldsymbol{\theta}}^{k+1} = \boldsymbol{\theta}^k - \frac{\alpha}{M} \sum_{m \in \mathcal{M}} \left( \boldsymbol{q}_m^{k-1} + \Delta \boldsymbol{q}_m^k \right). \qquad (7)$$

**Lemma 3.1.** *The influence of lazy aggregation at communication round $k$ can be bounded by*

$$\left\|\tilde{\boldsymbol{\theta}}^k - \boldsymbol{\theta}^k\right\|_2^2 \leqslant \frac{4\alpha^2 |\mathcal{M}_c^k|}{M^2} \sum_{m \in \mathcal{M}_c^k} \left( \left( \left\|\nabla f_m(\boldsymbol{\theta}^k) - \boldsymbol{q}_m^{k-1}\right\|_2 - \left\|\tau_m^k R_m^k \mathbf{1}\right\|_2 \right)^2 + 4(R_m^k)^2 d + \frac{d}{2} \right). \quad (8)$$

Corresponding to Lemma 3.1, since $R_m^k$ is independent of $\tau_m^k$, we can formulate an optimization problem to minimize the upper bound of this model deviation caused by update skipping for each device $m$:

$$\begin{aligned}
\underset{0 < \tau_m^k \leqslant 1}{\text{minimize}} \quad & \left( \left\|\nabla f_m(\boldsymbol{\theta}^k) - \boldsymbol{q}_m^{k-1}\right\|_2 - \left\|\tau_m^k R_m^k \mathbf{1}\right\|_2 \right)^2 \\
\text{subject to} \quad & \tau_m^k = \frac{1}{\left(2^{b_m^k} - 1\right)}
\end{aligned} \quad (9)$$

Solving the below optimization problem gives AQUILA an adaptive strategy (10) that selects the optimal quantization level based on the quantization range $R_m^k$, the dimension $d$ of the local model, the current gradient $\nabla f_m(\boldsymbol{\theta}^k)$, and the last uploaded quantized gradient $\boldsymbol{q}_m^{k-1}$:

$$(b_m^k)^* = \left\lfloor \log_2 \left( \frac{R_m^k \sqrt{d}}{\left\|\nabla f_m(\boldsymbol{\theta}^k) - \boldsymbol{q}_m^{k-1}\right\|_2} + 1 \right) \right\rfloor. \quad (10)$$

The superiority of (10) comes from the following three aspects. First, since $R_m^k \geqslant [\nabla f_m(\boldsymbol{\theta}^k)]_i - [\boldsymbol{q}_m^{k-1}]_i \geqslant -R_m^k$, the optimal quantization level $(b_m^k)^*$ must be greater than or equal to 1. Second, AQUILA can personalize an optimal quantization level for each device corresponding to its own gradient, whereas, in AdaQuantFL, each device merely utilizes an identical quantization level according to the global loss. Third, the gradient innovation and quantization range $R_m^k$ tend to fluctuate along with the training process instead of keeping descending, and thus prevent the quantization level from increasing tremendously compared with AdaQuantFL.

## 3.2 Precise lazy aggregation criterion

**Definition 3.3 (Quantization error).** *The global quantization error $\varepsilon^k$ is defined by the subtraction between the current unquantized gradient $\nabla f(\boldsymbol{\theta}^k)$ and its quantized value $\boldsymbol{q}^{k-1} + \Delta \boldsymbol{q}^k$, i.e.,*

$$\varepsilon^k = \nabla f(\boldsymbol{\theta}^k) - \boldsymbol{q}^{k-1} - \Delta \boldsymbol{q}^k, \quad (11)$$

*where $\nabla f(\boldsymbol{\theta}^k) = \sum_{m \in \mathcal{M}} \nabla f_m(\boldsymbol{\theta}^k), \boldsymbol{q}^{k-1} = \sum_{m \in \mathcal{M}} \boldsymbol{q}_m^{k-1}, \Delta \boldsymbol{q}^k = \sum_{m \in \mathcal{M}} \Delta \boldsymbol{q}_m^k$.*

To better fit the larger quantization errors induced by fewer quantization bits in (10), AQUILA possesses a new communication criterion to avoid the potential expansion of the devices group being skipped:

$$\left\|\Delta \boldsymbol{q}_m^k\right\|_2^2 + \left\|\varepsilon_m^k\right\|_2^2 \leqslant \frac{\beta}{\alpha^2} \left\|\boldsymbol{\theta}^k - \boldsymbol{\theta}^{k-1}\right\|_2^2, \forall m \in \mathcal{M}_c^k, \quad (12)$$

where $\beta \geqslant 0$ is a tuning factor. Note that this skipping rule is employed at epoch $k$, in which each device $m$ calculates its quantized gradient innovation $\Delta \boldsymbol{q}_m^k$ and quantization error $\varepsilon_m^k$, then utilizes this rule to decide whether uploads $\Delta \boldsymbol{q}_m^k$.

The comparison of AQUILA's skip rule and LAQ's is also shown in Appendix A.2. Instead of storing a large number of previous model parameters as LAQ, the strength of (12) is that AQUILA directly utilizes the global model for two adjacent rounds as the skip condition, which does not need to estimate the global gradient (more precise), requires fewer hyperparameters to adjust, and considerably reduces the storage pressure of local devices. This is especially important for small-capacity devices (e.g., sensors) in practical IoT scenarios.

---

**Algorithm 1** Communication Efficient FL with AQUILA

---

**Input:** the number of communication rounds $K$, the learning rate $\alpha$.
**Initialize:** the initial global model parameter $\boldsymbol{\theta}^0$.
 1: Server broadcasts $\boldsymbol{\theta}^0$ to all devices.                                                  ▷ For the initial round $k = 0$.
 2: **for** each device $m \in \mathcal{M}$ **in parallel do**
 3:     Calculates local gradient $\nabla f_m(\boldsymbol{\theta}^0)$.
 4:     Compute $(b_m^0)^*$ by setting $\boldsymbol{q}_m^{k-1} = \boldsymbol{0}$ in (10) and the quantized gradient innovation $\Delta \boldsymbol{q}_m^0$, and transmits it back to the server side.
 5: **end for**
 6: **for** $k = 1, 2, ..., K$ **do**
 7:     Server broadcasts $\boldsymbol{\theta}^k$ to all devices.
 8:     **for** each device $m \in \mathcal{M}$ **in parallel do**
 9:         Calculates local gradient $\nabla f_m(\boldsymbol{\theta}^k)$, the optimal local quantization level $(b_m^k)^*$ by (10), and the quantized gradient innovation $\Delta \boldsymbol{q}_m^k$.
10:         **if** (12) does not hold for device $m$ **then**                 ▷ If satisfies, skip uploading.
11:             device $m$ transmits $\Delta \boldsymbol{q}_m^k$ to the server.
12:         **end if**
13:     **end for**
14:     Server updates $\boldsymbol{\theta}^{k+1}$ by the saving previous global quantized gradient $\boldsymbol{q}_m^{k-1}$ and the received quantized gradient innovation $\Delta \boldsymbol{q}_m^k$: $\boldsymbol{\theta}^{k+1} := \boldsymbol{\theta}^k - \alpha \left( \boldsymbol{q}^{k-1} + 1/M \sum_{m \in \mathcal{M}^k} \Delta \boldsymbol{q}_m^k \right)$.
15:     Server saves the average quantized gradient $\boldsymbol{q}^k$ for the next aggregation.
16: **end for**

---

The detailed process of AQUILA is comprehensively summarized in Algorithm 1. At epoch $k = 0$, each device calculates $b_m^0$ by setting $\boldsymbol{q}_0^{k-1} = \boldsymbol{0}$ and uploads $\Delta \boldsymbol{q}_0^k$ to the server since the (12) is not satisfied. At epoch $k \in \{1, 2, ..., K\}$, the server first broadcasts the global model $\boldsymbol{\theta}^k$ to all devices. Each device $m$ computes $\nabla f(\boldsymbol{\theta}_m^k)$ with local training data and then utilizes it to calculate an optimal quantization level by (10). Subsequently, each device computes its gradient innovation after quantization and determines whether or not to upload based on the communication criterion (12). Finally, the server updates the new global model $\boldsymbol{\theta}^{k+1}$ with up-to-date quantized gradients $\boldsymbol{q}_m^{k-1} + \Delta \boldsymbol{q}_m^k$ for those devices who transmit the uploads at epoch $k$, while reusing the old quantized gradients $\boldsymbol{q}_m^{k-1}$ for those who skip the uploads.

## 4 THEORETICAL DERIVATION AND ANALYSIS OF AQUILA

As aforementioned, we bound the model deviation caused by skipping updates with respect to quantization bits. Specifically, if the communication criterion (12) holds for the device $m$ at epoch $k$, it does not contribute to epoch $k$'s gradient. Otherwise, the loss caused by device $m$ will be minimized with the optimal quantization level selection criterion (10). In this section, the theoretical convergence derivation of AQUILA is based on the following standard assumptions.

**Assumption 4.1 (L-smoothness).** *Each local objective function $f_m$ is $L_m$-smooth, i.e., there exist a constant $L_m > 0$, such that $\forall \boldsymbol{x}, \boldsymbol{y} \in \mathbb{R}^d$,*

$$\|\nabla f_m(\mathbf{x}) - \nabla f_m(\mathbf{y})\|_2 \leqslant L_m \|\mathbf{x} - \mathbf{y}\|_2, \tag{13}$$

*which implies that the global objective function $f$ is $L$-smooth with $L \leq \bar{L} = \frac{1}{m} \sum_{i=1}^m L_m$.*

**Assumption 4.2 (Uniform lower bound).** *For all $\boldsymbol{x} \in \mathbb{R}^d$, there exist $f^* \in \mathbb{R}$ such that $f(\boldsymbol{x}) \geq f^*$.*

**Lemma 4.1.** *Following the assumption that the function $f$ is $L$-smooth, we have*

$$f(\boldsymbol{\theta}^{k+1}) - f(\boldsymbol{\theta}^k) \leqslant -\frac{\alpha}{2} \left\| \nabla f(\boldsymbol{\theta}^k) \right\|_2^2 + \alpha \left( \left\| \frac{1}{M} \sum_{m \in \mathcal{M}_c^k} \Delta \boldsymbol{q}_m^k \right\|_2^2 + \left\| \boldsymbol{\varepsilon}^k \right\|_2^2 \right) + \left( \frac{L}{2} - \frac{1}{2\alpha} \right) \left\| \boldsymbol{\theta}^{k+1} - \boldsymbol{\theta}^k \right\|_2^2.$$

$$\tag{14}$$

## 4.1 CONVERGENCE ANALYSIS FOR GENERALLY NON-CONVEX CASE.

**Theorem 4.1.** *Suppose Assumptions 4.1, 4.2, and B.1 (29) be satisfied. If $\mathcal{M}_c^k \neq \varnothing$, the global objective function $f$ satisfies*

$$f(\boldsymbol{\theta}^{k+1}) - f(\boldsymbol{\theta}^k) \leqslant -\frac{\alpha}{2} \left\| \nabla f(\boldsymbol{\theta}^k) \right\|_2^2 + \left( \frac{L}{2} - \frac{1}{2\alpha} \right) \left\| \boldsymbol{\theta}^{k+1} - \boldsymbol{\theta}^k \right\|_2^2 + \frac{\beta\gamma}{\alpha} \left\| \boldsymbol{\theta}^k - \boldsymbol{\theta}^{k-1} \right\|_2^2. \tag{15}$$

**Corollary 4.1.** *Let all the assumptions of Theorem 4.1 hold and $\frac{L}{2} - \frac{1}{2\alpha} + \frac{\beta\gamma}{\alpha} \leqslant 0$, then the AQUILA requires*

$$K = \mathcal{O} \left( \frac{2\omega_1}{\alpha\epsilon^2} \right) \tag{16}$$

*communication rounds with $\omega_1 = f\left(\boldsymbol{\theta}^1\right) - f\left(\boldsymbol{\theta}^*\right) + \frac{\beta\gamma}{\alpha} \left\| \boldsymbol{\theta}^1 - \boldsymbol{\theta}^0 \right\|_2^2$ to achieve $\min_k \|\nabla f(\boldsymbol{\theta}^k)\|_2^2 \leqslant \epsilon^2$.*

**Compared to LAG.** Corresponding to Eq.(70) in Chen et al. (2018), LAG defines a Lyapunov function $\mathbb{V}^k := f(\boldsymbol{\theta}^k) - f(\boldsymbol{\theta}^*) + \sum_{d=1}^D \beta_d \|\boldsymbol{\theta}^{k+1-d} - \boldsymbol{\theta}^{k-d}\|_2^2$ and claims that it satisfies

$$\mathbb{V}^{k+1} - \mathbb{V}^k \leq -\left( \frac{\alpha}{2} - \tilde{c}\left(\alpha, \beta_1\right)(1+\rho)\alpha^2 \right) \left\| \nabla f(\boldsymbol{\theta}^k) \right\|_2^2, \tag{17}$$

where $\tilde{c}(\alpha, \beta_1) = L/2 - 1/(2\alpha) + \beta_1$, $\beta_1 = D\xi/(2\alpha\eta)$, $\xi < 1/D$, and $\rho > 0$. The above result (17) indicates that LAG requires

$$K_{LAG} = \mathcal{O} \left( \frac{2\omega_1}{(\alpha - 2\tilde{c}\left(\alpha, \beta_1\right)(1+\rho)\alpha^2)\epsilon^2} \right) \tag{18}$$

communication rounds to converge. Since the non-negativity of the term $\tilde{c}(\alpha, \beta_1)(1+\rho)\alpha^2$, we can readily derive that $\alpha < \alpha - 2\tilde{c}(\alpha, \beta_1)(1+\rho)\alpha^2$, which demonstrates AQUILA achieves a better convergence rate than LAG with the appropriate selection of $\alpha$.

## 4.2 CONVERGENCE ANALYSIS UNDER POLYAK-ŁOJASIEWICZ CONDITION.

**Assumption 4.3 ($\mu-$PŁ condition).** *Function $f$ satisfies the PL condition with a constant $\mu > 0$, that is,*

$$\left\| \nabla f(\boldsymbol{\theta}^k) \right\|_2^2 \geqslant 2\mu(f(\boldsymbol{\theta}^k) - f(\boldsymbol{\theta}^*)). \tag{19}$$

**Theorem 4.2.** *Suppose Assumptions 4.1, 4.2, and 4.3 be satisfied and $\mathcal{M}_c^k \neq \varnothing$, if the hyperparameters satisfy $\frac{\beta\gamma}{\alpha} \leqslant (1 - \alpha\mu)\left(\frac{1}{2\alpha} - \frac{L}{2}\right)$, then the global objective function satisfies*

$$f(\boldsymbol{\theta}^{k+1}) - f(\boldsymbol{\theta}^k) \leqslant -\alpha\mu(f(\boldsymbol{\theta}^k) - f(\boldsymbol{\theta}^*)) + \left( \frac{L}{2} - \frac{1}{2\alpha} \right) \left\| \boldsymbol{\theta}^{k+1} - \boldsymbol{\theta}^k \right\|_2^2 + \frac{\beta\gamma}{\alpha} \left\| \boldsymbol{\theta}^k - \boldsymbol{\theta}^{k-1} \right\|_2^2, \tag{20}$$

*and the AQUILA requires*

$$K = \mathcal{O} \left( -\frac{1}{\log(1-\alpha\mu)} \log \frac{\omega_1}{\epsilon} \right) \tag{21}$$

*communication round with $\omega_1 = f(\boldsymbol{\theta}^1) - f(\boldsymbol{\theta}^*) + \left(\frac{1}{2\alpha} - \frac{L}{2}\right) \left\| \boldsymbol{\theta}^1 - \boldsymbol{\theta}^0 \right\|_2^2$ to achieve $f(\boldsymbol{\theta}^{K+1}) - f(\boldsymbol{\theta}^*) + (\frac{1}{2\alpha} - \frac{L}{2})\|\boldsymbol{\theta}^{K+1} - \boldsymbol{\theta}^K\|_2^2 \leqslant \epsilon$.*

**Compared to LAG.** According to Eq.(50) in Chen et al. (2018), we have that

$$\mathbb{V}^K \leq \left( 1 - \alpha\mu + \alpha\mu\sqrt{D\xi} \right)^K \mathbb{V}^0, \tag{22}$$

where $\xi < 1/D$. Thus, we have that LAG requires

$$K_{LAG} = \mathcal{O} \left( -\frac{1}{\log(1-\alpha\mu+\alpha\mu\sqrt{D\xi})} \log \frac{\omega_1}{\epsilon} \right) \tag{23}$$

communication rounds to converge. Compared to Theorem 4.2, we can derive that $\log(1 - \alpha\mu) < \log(1 - \alpha\mu + \alpha\mu\sqrt{D\xi})$, which indicates that AQUILA has a faster convergence than LAG under the PŁ condition.

**Remark.** We want to emphasize that LAQ introduces the Lyapunov function into its proof, making it extremely complicated. In addition, LAQ can only guarantee that the final objective function converges to a range of the optimal solution rather than an accurate optimum $f(\boldsymbol{\theta}^*)$. Nevertheless, as discussed in Section 3.2, we utilize the precise model difference in AQUILA as a surrogate for the global gradient and thus simplify the proof.

## 5 EXPERIMENTS AND DISCUSSION

### 5.1 EXPERIMENT SETUP

In this paper, we evaluate AQUILA on `CIFAR-10`, `CIFAR-100` (Krizhevsky et al., 2009), and `WikiText-2` dataset (Merity et al., 2016), considering IID, Non-IID data scenario, and heterogeneous model architecture (which is also a crucial challenge in FL) simultaneously.

The FL environment is simulated in Python 3.9 with `PyTorch 11.1` (Paszke et al., 2019) implementation. For the diversity of the neural network structures, we train `ResNet-18` (He et al., 2016) at `CIFAR-10` dataset, `MobileNet-v2` (Sandler et al., 2018) at `CIFAR-100` dataset, and `Transformer` (Vaswani et al., 2017) at `WikiText-2` dataset.

As for the FL system setting, in the majority of our experiments, the whole system exists $M = 10$ total devices. However, considering the large-scale feature of FL, we also validate AQUILA on a larger system of $M = 100/80$ total devices for `CIFAR` / `WikiText-2` dataset. The hyperparameters and additional details of our experiments are revealed in Appendix A.3.

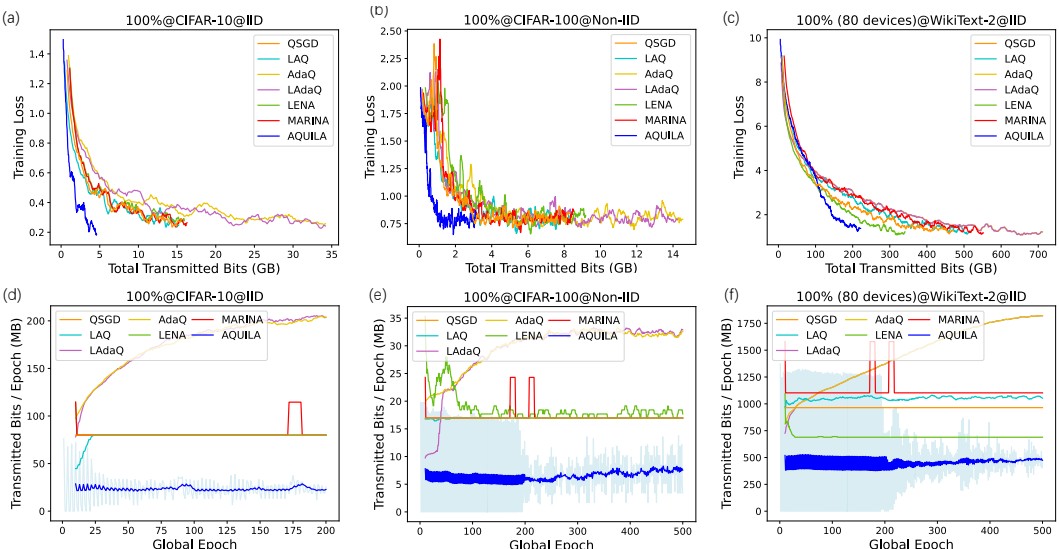

Figure 1: Comparison of AQUILA with other communication-efficient algorithms on IID and Non-IID settings with **homogeneous** model structure. (a)-(c): training loss v.s. total transmitted bits, (d)-(f): transmitted bits per epoch v.s. global epoch.

### 5.2 HOMOGENEOUS ENVIRONMENT

We first evaluate AQUILA with homogeneous settings where all local models share the same model architecture as the global model. To better demonstrate the effectiveness of AQUILA, its performance is compared with several state-of-the-art methods, including AdaQuantFL, LAQ with fixed levels, LENA (Ghadikolaei et al., 2021), MARINA (Gorbunov et al., 2021), and the naive combination of AdaQuantFL with LAQ. Note that based on this homogeneous setting, we conduct both IID and Non-IID evaluations on `CIFAR-10` and `CIFAR-100` dataset, and an IID evaluation on `WikiText-2`. To simulate the Non-IID FL setting as (Diao et al., 2020), each device is allocated two classes of data in `CIFAR-10` and 10 classes of data in `CIFAR-100` at most, and the amount of data for each label is balanced.

The experimental results are presented in Fig. 1, where *100%* implies all local models share a similar structure with the global model (i.e., homogeneity), *100% (80 devices)* denotes the experiment is conducted in an 80 devices system, and *LAdaQ* represents the naive combination of AdaQuantFL and LAQ. For better illustration, the results have been smoothed by their standard deviation. The solid lines represent values after smoothing, and transparent shades of the same colors around them represent the true values. Additionally, Table 2 shows the total number of bits transmitted by all devices throughout the FL training process. The comprehensive experimental results are established in Appendix A.4.

## 5.3 NON-HOMOGENEOUS SCENARIO

In this section, we also evaluate AQUILA with heterogeneous model structures as HeteroFL (Diao et al., 2020), where the structures of local models trained on the device side are heterogeneous. Suppose the global model at epoch $k$ is $\boldsymbol{\theta}^k$ and its size is $d = w_g * h_g$, then the local model of each device $m$ can be selected by $\boldsymbol{\theta}_m^k = \boldsymbol{\theta}^k [: w_m, : h_m]$, where $w_m = r_m w_g$ and $h_m = r_m h_g$, respectively. In this paper, we choose model complexity levels $r_m = 0.5$.

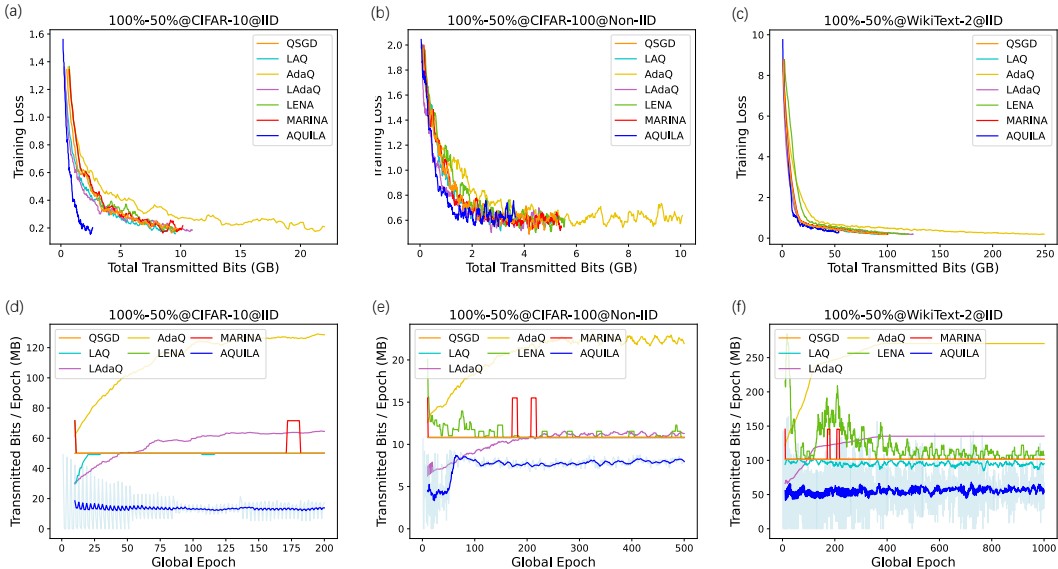

Figure 2: Comparison of AQUILA with other communication-efficient algorithms on IID and Non-IID settings with **heterogeneous** model structure. (a)-(c): training loss v.s. total transmitted bits, (d)-(f): transmitted bits per epoch v.s. global epoch.

Most of the symbols in Fig. 2 are identical to the Fig. 1. *100%-50%* is a newly introduced symbol that implies half of the devices share the same structure with the global model while another half only have 50% * 50% parameters as the global model.

**Performance Analysis.** First of all, AQUILA achieves a significant transmission reduction compared to the naive combination of LAQ and AdaQuantFL in all datasets, which demonstrates the superiority of AQUILA's efficiency. Specifically, Table 2 indicates that AQUILA saves 57.49% of transmitted bits in the system of 80 devices at the `WikiText-2` dataset and reduces 23.08% of transmitted bits in the system of 100 devices at the `CIFAR-100` dataset, compared to the naive combination. And other results in Table 3 also show an obvious reduction in terms of the total transmitted bits required for convergence.

Second, in Fig. 1 and Fig. 2, the changing trend of AQUILA's communication bits per each round clearly verifies the necessity and effectiveness of our well-designed adaptive quantization level and skip criterion. In these two figures, the number of bits transmitted in each round of AQUILA **fluctuates** a bit, indicating the effectiveness of AQUILA's selection rule. Meanwhile, the value of transmitted bits remains at quite a low level, suggesting that the adaptive quantization principle makes training more efficient. Moreover, the figures also inform that the quantization level selected by

AQUILA will not continuously increase during training instead of being as AdaQuantFL. In addition, based on these two figures, we can also conclude that AQUILA converges faster under the same communication costs.

Finally, AQUILA is capable of adapting to a wide range of challenging FL circumstances. In the Non-IID scenario and heterogeneous model structure, AQUILA still outperforms other algorithms by significantly reducing overall transmitted bits while maintaining the same convergence property and objective function value. In particular, AQUILA reduces 60.4% overall communication costs compared to LENA and 57.2% compared to MARINA on average. These experimental results in non-homogeneous FL settings prove that AQUILA can be stably employed in more general and complicated FL scenarios.

### 5.4 ABLATION STUDY ON THE IMPACT OF TUNING FACTOR $\beta$

One key contribution of AQUILA is presenting a new lazy aggregation criterion (12) to reduce communication frequency. In this part, we evaluate the effects of the loss performance of different tuning factor $\beta$ value in Fig. 3. As $\beta$ grows within a certain range, the convergence speed of the model will slow down (due to lazy aggregation). Still, it will eventually converge to the same model performance while considerably reducing the communication overhead. Nevertheless, increasing the value of $\beta$ will lead to a decrease in the final model performance since it skips so many essential uploads that make the training deficient. The accuracy (perplexity) comparison of AQUILA with various selections of the tuning factor $\beta$ is shown in Fig. 10, which indicates the same trend.To sum up, we should choose the value of factor $\beta$ to maintain the model's performance and minimize the total transmitted amount of bits. Specifically, we select the value of $\beta = 0.1, 0.25, 1.25$ for `CIFAR-10`, `CIFAR-100`, and `WikiText-2` datasets for our evaluation, respectively.

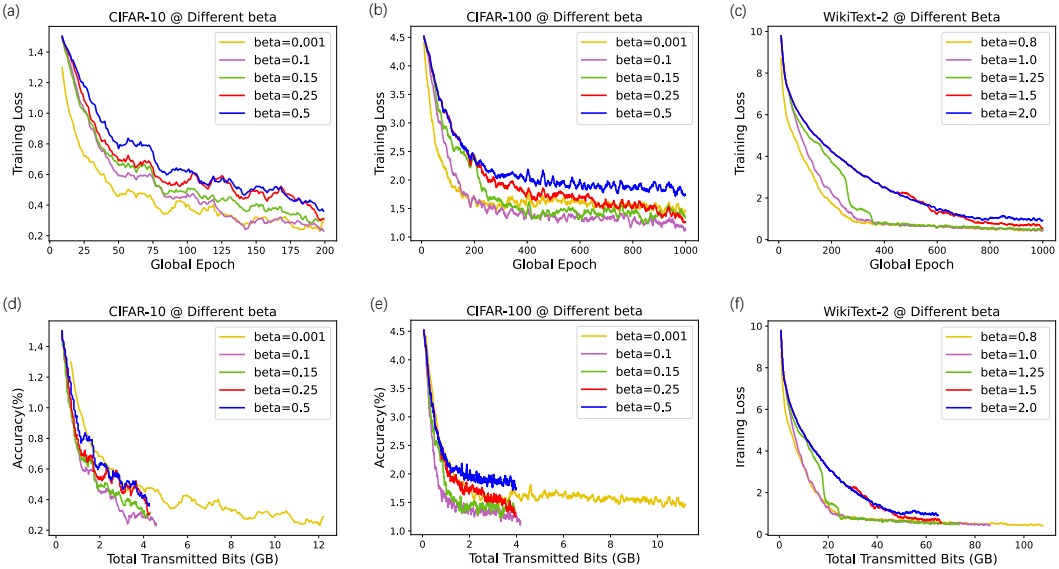

Figure 3: Comparison of AQUILA with various selections of the tuning factor $\beta$ in three datasets.

## 6 CONCLUSIONS AND FUTURE WORK

This paper proposes a communication-efficient FL procedure to simultaneously adjust two mutually-dependent degrees of freedom: communication frequency and quantization level. With the close cooperation of the novel adaptive quantization and adjusted lazy aggregation strategy derived in this paper, the proposed AQUILA has been proven to be capable of reducing the transmitted costs while maintaining the convergence guarantee and model performance compared to existing methods. The evaluation with Non-IID data distribution and various heterogeneous model architectures demonstrates that AQUILA is compatible in a non-homogeneous FL environment.

REPRODUCIBILITY

We present the overall theorem statements and proofs for our main results in the Appendix, as well as necessary experimental plotting figures. Furthermore, we submit the code of AQUILA in the supplementary material part, including all the hyperparameters and a *requirements* to help the public reproduce our experimental results. Our algorithm is straightforward, well-described, and easy to implement.

ETHICS STATEMENT

All evaluations of AQUILA are performed on publicly available datasets for reproducibility purposes. This paper empirically studies the performance of various state-of-art algorithms, therefore, probably introduces no new ethical or cultural problems. This paper does not utilize any new dataset.

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

# A APPENDIX

The appendix includes supplementary experimental results, mathematical proof of the aforementioned theorems, and a detailed derivation of the novel adaptive quantization criterion and lazy aggregation strategy. Compared to Fig. 1 and Fig. 2 in the main text, the result figures in the appendix show a more comprehensive evaluation with AQUILA, which contains more detailed information including but not limited to *accuracy vs steps* and *training loss vs steps* curves.

## A.1 OVERALL FRAMEWORK OF AQUILA

The cooperation of the novel adaptive quantization criterion (10) and lazy aggregation strategy (12) is illustrated in Fig. 4a. Compared to the naive combination of AdaQuantFL and LAQ, where the mutual influence between adaptive quantization and lazy aggregation has not been considered, as shown in Fig. 4b, AQUILA adaptively optimizes the allocation of quantization bits throughout training to promote the convergence of lazy aggregation, and at the same time utilizes the lazy aggregation strategy to improve the efficiency of adaptive quantization by compress the transmission with a lower quantization level.

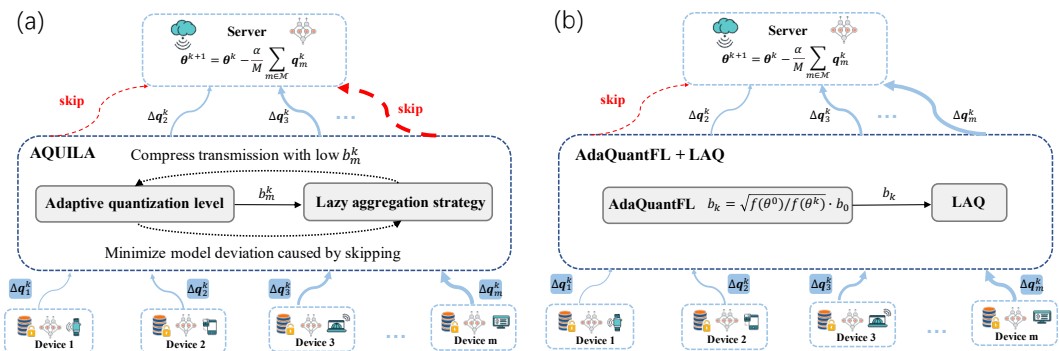

Figure 4: The schematic illustration of the communication-efficient FL with AQUILA in comparison with the naive combination of AdaQuantFL and LAQ. The blue lines indicating the transmission of quantized gradient innovation in AQUILA are drawn in different thicknesses to represent various sizes of quantized gradient innovation, considering the heterogeneous FL environment as in our evaluation part. For instance, the gradient innovation $\Delta q_3^k$ of a PC is larger than $\Delta q_2^k$ of a mobile phone.

## A.2 EXPLANATION OF THE QUANTIZER AND THE SKIP RULE OF LAQ'S

The quantizer (6) is a deterministic quantizer that, at each dimension, maps the gradient innovation to the closest point at a one-dimensional grid. The range of the grid is $R_m^k$, and the granularity is determined by quantization level $\tau_m^k$. Each dimension of gradient innovation is mapped to an integer in $\{0, 1, 2, 3, \ldots, 2^b - 1\}$. More precisely, the $1/2$ ensures mapping to the closest integer instead of flooring to a smaller integer. The $R_m^k$ in the numerator ensures that the mapped integer is non-negative. As a result, when the gradient innovation is transmitted to the central server, 32 bits are used for the range, and $b * d$ bits are used for the mapped integer. Thus, $32 + b * d$ bits are transmitted in total.

The difference between (6) and (32) (Lemma B.2) is that (6) encodes the raw gradient innovation vector to an integer vector, whilst (32) decodes the integer vector to a quantized gradient innovation vector. Specifically, in the training process, each client utilizes (6) to encode the gradient innovation to an integer at each dimension, and afterwards, the integer vector $\psi_m^k$ and $\tau_m^k$ are sent to a central server. After receiving them, the central server can decode the quantized gradient innovation as (32) states.

The skip rule of LAQ is measured by the summation of the accumulated model difference and quantization error:

$$\|\Delta \boldsymbol{q}_m^k\|_2^2 \leqslant \frac{1}{\alpha^2 M^2} \sum_{d'=1}^{D} \xi_{d'} \left\|\boldsymbol{\theta}^{k+1-d'} - \boldsymbol{\theta}^{k-d'}\right\|_2^2 + 3 \left(\left\|\boldsymbol{\varepsilon}_m^k\right\|_2^2 + \left\|\hat{\boldsymbol{\varepsilon}}_m^{k-1}\right\|_2^2\right), \qquad (24)$$

where $\xi_{d'}$ is a series of manually selected scalars and $D$ is also predetermined. $\varepsilon_m^k$ is the quantization error of client m at epoch k, and $\hat{\varepsilon}_m^{k-1}$ is the quantization of client m at last time it uploads its gradient innovation. Please refer to Sun et al. (2020) for more details on (24). In order to compute the LAQ skip threshold, each client has to store enormous previous information.

The difference of AQUILA skipping criterion and LAQ skipping criterion is as follows. First, the AQUILA threshold is easier to compute for a local client. Compared to the LAQ skipping criterion, AQUILA skipping criterion is more concise and thus requires less storage and computing power. Second, the AQUILA criterion is easier to tune because much fewer hyperparameters are introduced. Compared to the LAQ criterion in which $\alpha$, $D$ and $\{\xi_{d'}\}_{d'=1}^{D}$ are all manually selected, whilst only two hyperparameters $\alpha$ and $\beta$ are introduced in the AQUILA criterion. Third, with the given threshold, AQUILA has a good theoretical property. The theoretical analysis of AQUILA is easier to follow with no Lyapunov function introduced as in LAQ. And the result also shows that AQUILA can achieve a better convergence rate under the non-convex case and the PL condition.

## A.3    EXPERIMENT SETUP

In this section, we provide some extra hyperparameter settings for our evaluation. For the LAQ, we set $D = 10$ and $\xi_1 = \xi_2 = \cdots = \xi_D = 0.8/D$ as the same as the setting in their paper. For LENA, we set $\beta_{LENA} = 40$ in their trigger condition. And for MARINA, we calculate the uploading probability of Bernoulli distribution as $p = \xi_Q/d$ as announced in their paper. In addition, we choose the CrossEntropy function as our objective function in the experiment part. Table 1 shows the hyperparameter details of our evaluation.

Table 1: The hyperparameters for `CIFAR-10`, `CIFAR-100`, and `WikiText-2` datasets in the FL training process.

| Dataset | `CIFAR-10` | | `CIFAR-100` | | `WikiText-2` |
|---|---|---|---|---|---|
| Model | ResNet-18 | | MobileNet-v2 | | Transformer |
| Data Distribution | IID | Non-IID | IID | Non-IID | IID |
| Global Epoch | 200 | 200 | 1000 | 500 | 1000 |
| Local Batch Size | 256 | 256 | 256 | 256 | 64 |
| Optimizer | SGD | SGD | SGD | SGD | SGD |
| Momentum | 0.9 | 0.9 | 0.9 | 0.9 | 0.9 |
| Weight Decay | 5.00E-04 | 5.00E-04 | 5.00E-04 | 5.00E-04 | 5.00E-04 |
| Learning Rate $\alpha$ | 0.1 | 0.1 | 0.1 | 0.1 | 0.5 |
| Tuning factor $\beta$ | 0.1 | 0.005 | 0.25 | 0.003 | 1.25 |

## A.4    COMPREHENSIVE EXPERIMENT RESULTS

This section will cover all the experimental results in our paper.

Table 2: Numerical numbers of total communication bits in the **homogeneous** environment.

| Total Comm Bits (GB) | | QSGD | AdaQ | LAQ | LAdaQ | LENA | MARINA | **AQUILA** |
|---|---|---|---|---|---|---|---|---|
| CIFAR-10 | IID-100 | 156.07 | 226.33 | 153.26 | 226.36 | 160.2 | 162.84 | **138.35** |
| | IID | 15.61 | 34.19 | 15.22 | 34.18 | 15.95 | 16.28 | **4.59** |
| | Non-IID | 15.61 | 20.39 | 14.48 | 19.86 | 17.64 | 16.28 | **11.53** |
| CIFAR-100 | IID-100 | 165.55 | 224.02 | 164.11 | 223.64 | 166.87 | 167.71 | **142.55** |
| | IID | 16.56 | 28.68 | 16.28 | 14.41 | 16.63 | 16.77 | **3.98** |
| | Non-IID | 8.28 | 14.54 | 8.27 | 14.25 | 9.19 | 8.49 | **6.12** |
| WikiText-2 | IID-80 | 470.95 | 711.49 | 513.07 | 710.17 | 341.17 | 338.38 | **218.59** |
| | IID | 134.56 | 340.97 | 106.92 | 170.40 | 150.07 | 136.31 | **71.91** |

Table 3: Numerical numbers of total communication bits in the **heterogeneous** environment.

| Total Comm Bits (GB) | | QSGD | Ada | LAQ | Ada+LAQ | LENA | MARINA | **AQUILA** |
|---|---|---|---|---|---|---|---|---|
| CIFAR-10 | IID | 9.76 | 21.99 | 9.55 | 10.98 | 9.97 | 10.18 | **2.65** |
| | Non-IID | 9.76 | 16.15 | 9.25 | 14.67 | 11.19 | 10.18 | **7.16** |
| CIFAR-100 | IID | 10.56 | 19.42 | 10.56 | 9.7 | 10.61 | 10.7 | **2.51** |
| | Non-IID | 5.28 | 10.07 | 5.28 | 5.02 | 5.56 | 5.42 | **3.66** |
| WikiText-2 | IID | 99.09 | 248.87 | 92.74 | 124.47 | 119.83 | 100.38 | **53.84** |

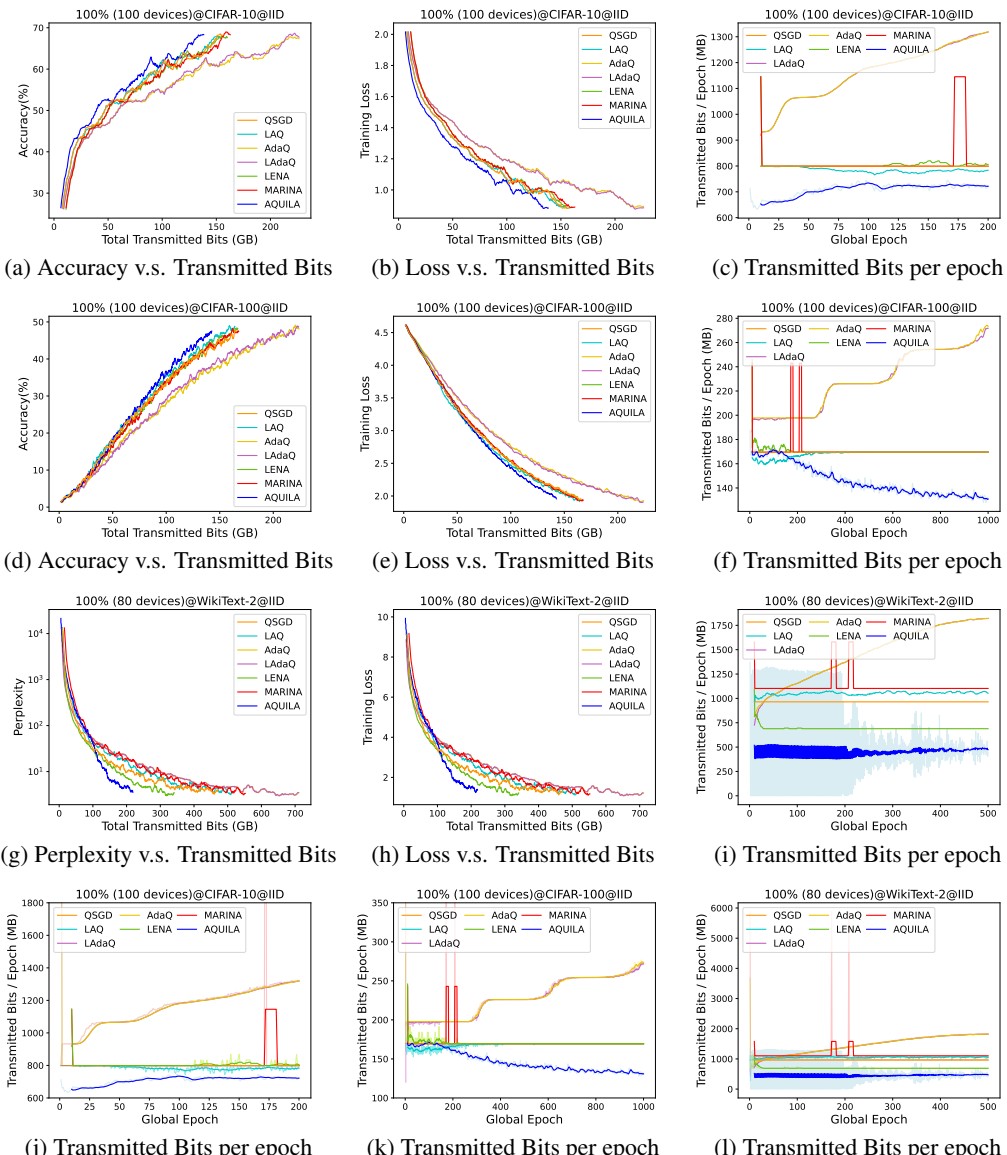

Figure 5: Comparison on 100 devices at `CIFAR-10` and `CIFAR-100`, 80 devices at `WikiText-2` dataset in the IID, homogeneous scenario.

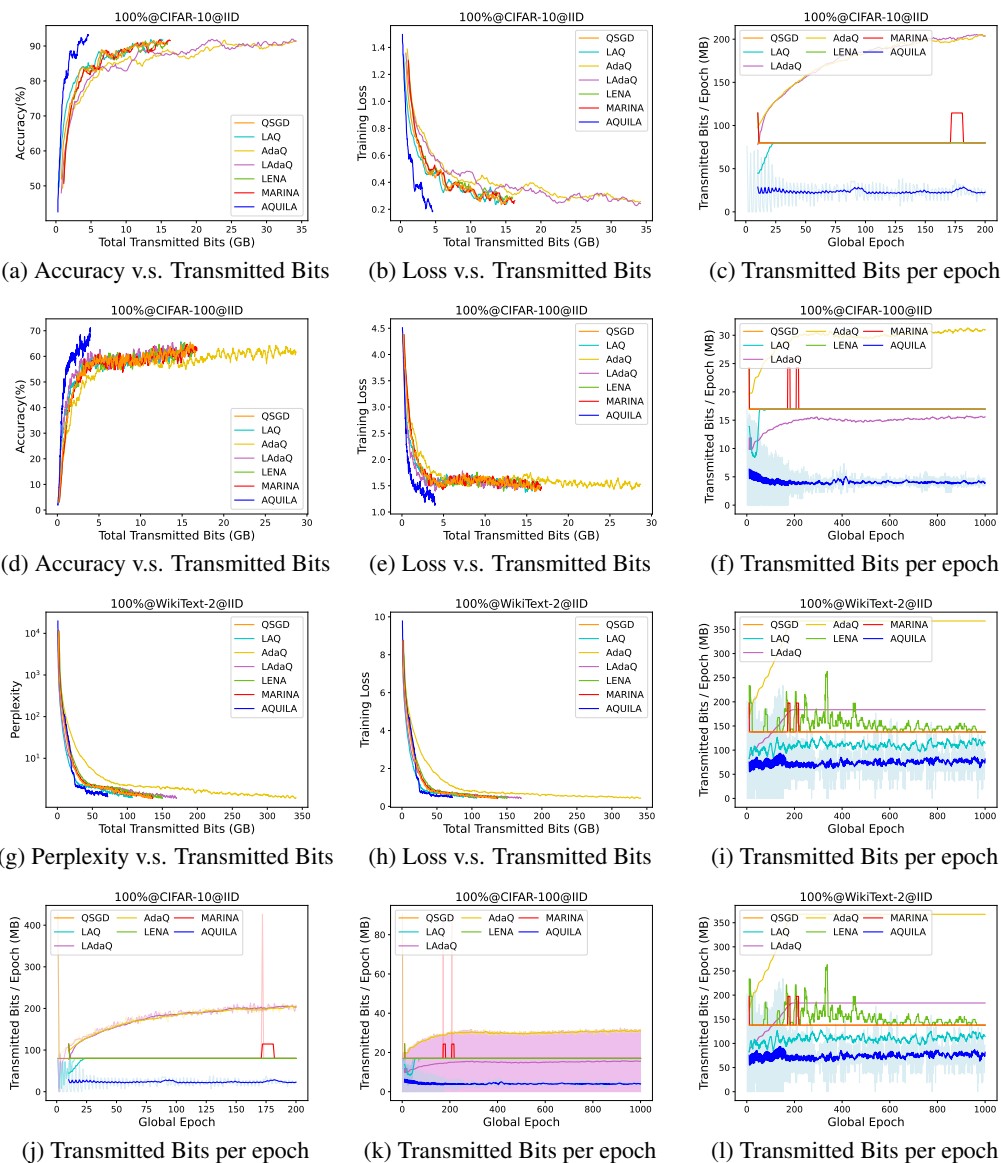

Figure 6: Comparison on 10 devices at three datasets in the IID, homogeneous scenario.

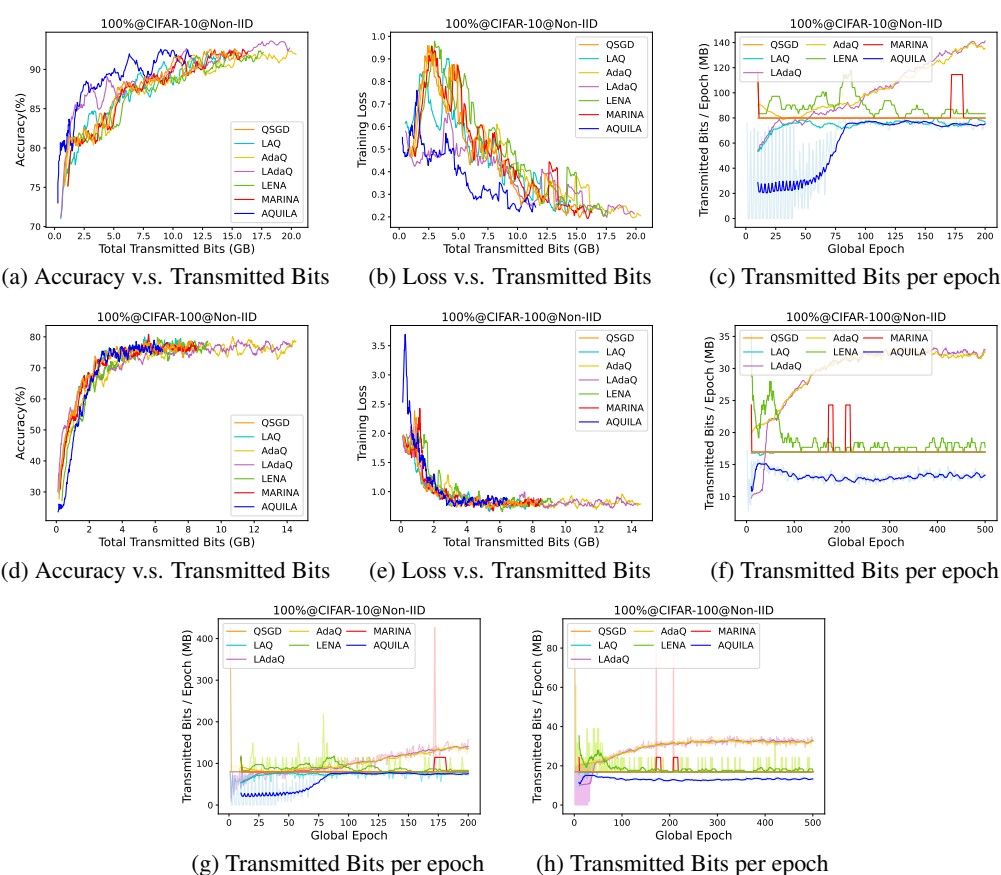

(a) Accuracy v.s. Transmitted Bits

(b) Loss v.s. Transmitted Bits

(c) Transmitted Bits per epoch

(d) Accuracy v.s. Transmitted Bits

(e) Loss v.s. Transmitted Bits

(f) Transmitted Bits per epoch

(g) Transmitted Bits per epoch

(h) Transmitted Bits per epoch

Figure 7: Comparison on 10 devices at three datasets in the Non-IID, homogeneous scenario..

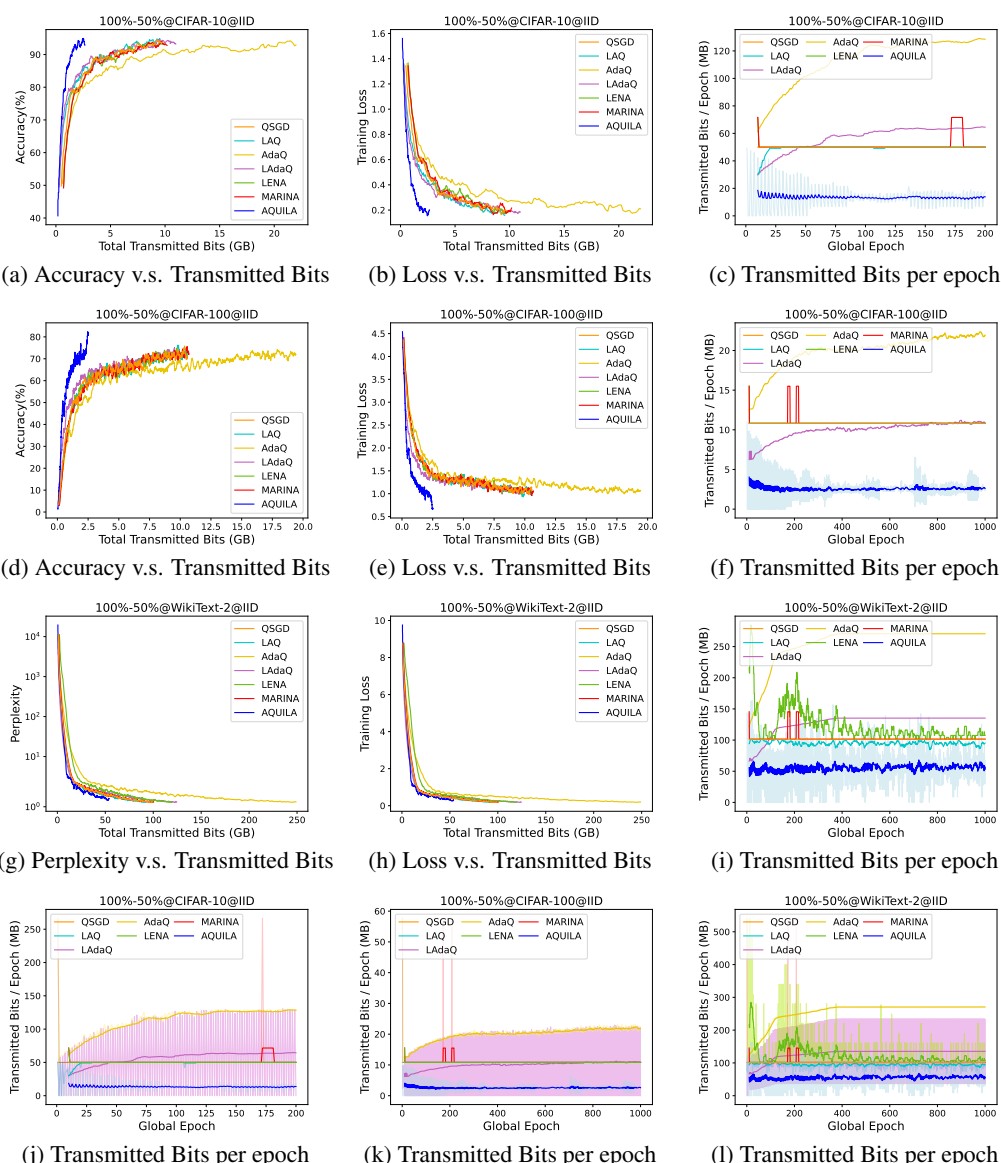

Figure 8: Comparison on 10 devices at three datasets in the IID, heterogeneous scenario.

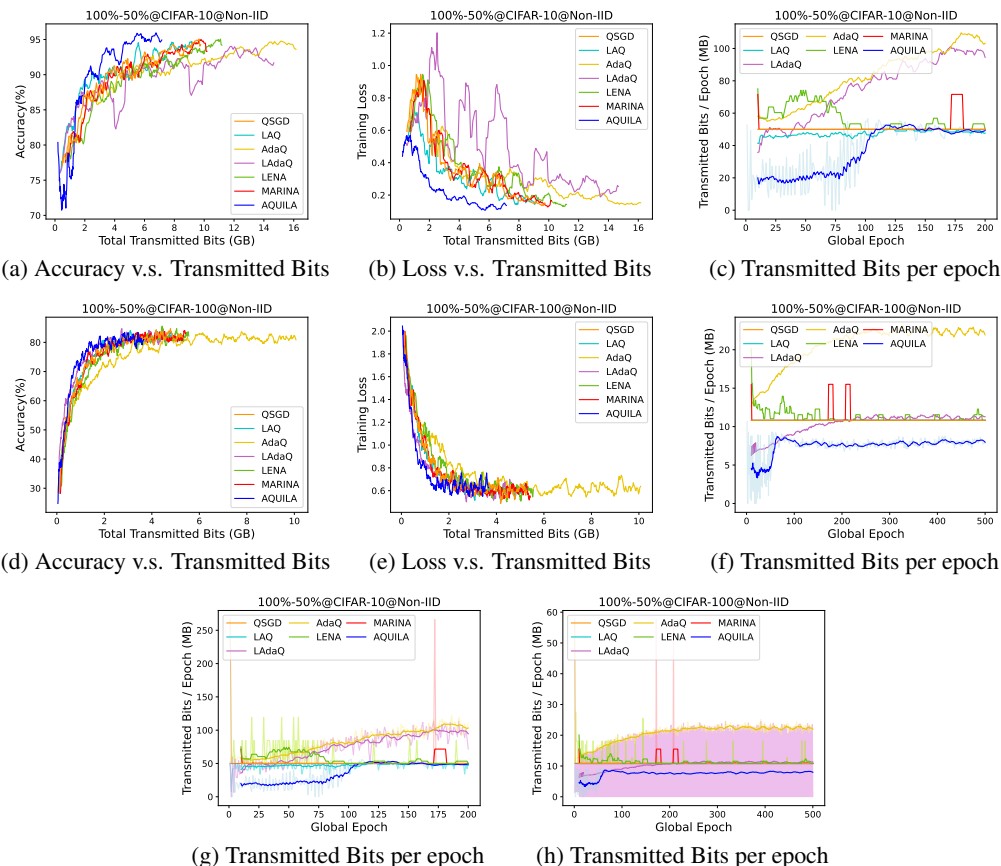

Figure 9: Comparison on 10 devices at three datasets in the Non-IID, heterogeneous scenario.

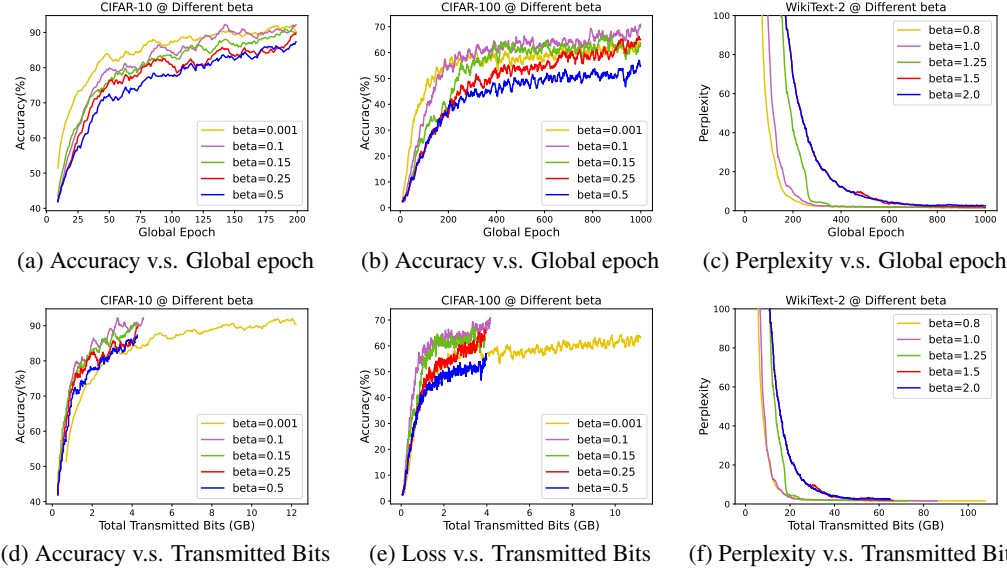

Figure 10: Accuracy (Perplexity) comparison of AQUILA with various selections of the tuning factor $\beta$ in three datasets.

# B   BASIC FACTS AND SOME LEMMAS

*Notations:* Bold fonts denote vectors (e.g., $\boldsymbol{\theta}$). Normal fonts denote scalars (e.g., $\alpha$). Subscript $m$ is used to describe functions about a local device $m$ (e.g., $f_m(\boldsymbol{\theta})$). A function without a subscript is used to describe an average among all devices (e.g., $f(\boldsymbol{\theta})$).

**Frequently used norm inequalities** Suppose $n \in \mathbb{N}^+$ and $\|\cdot\|_2$ denotes the $\ell^2-$norm. For $p$ in $\mathbb{R}^+, \boldsymbol{x}_i, \boldsymbol{a}, \boldsymbol{b} \in \mathbb{R}^d$, there holds

1. **Norm summation inequality.**

$$\left\| \sum_{i=1}^n \boldsymbol{x}_i \right\|_2^2 \leqslant n \sum_{i=1}^n \|\boldsymbol{x}_i\|_2^2 . \tag{25}$$

2. **Inner-product inequality.**

$$\langle \boldsymbol{a}, \boldsymbol{b} \rangle = \frac{1}{2} \left( \|\boldsymbol{a}\|_2^2 + \|\boldsymbol{b}\|_2^2 - \|\boldsymbol{a} - \boldsymbol{b}\|_2^2 \right) . \tag{26}$$

3. **Young's Inequality.**

$$\|\boldsymbol{a} + \boldsymbol{b}\|_2^2 \leqslant (1 + p) \|\boldsymbol{a}\|_2^2 + (1 + p^{-1}) \|\boldsymbol{b}\|_2^2 . \tag{27}$$

4. **Minkowski's Inequality.**

$$\|\boldsymbol{a} + \boldsymbol{b}\|_2 \leqslant \|\boldsymbol{a}\|_2 + \|\boldsymbol{b}\|_2 . \tag{28}$$

**Assumption B.1.** *All devices' quantization errors $\boldsymbol{\varepsilon}^k$ will be constrained by the total error of the omitted devices., i.e., $\forall\, k = 0, 1, \cdots, K$, if $\mathcal{M}_c^k \neq \varnothing$, $\exists\, \gamma \geqslant 1$, such that*

$$\|\boldsymbol{\varepsilon}^k\|_2^2 = \left\| \frac{1}{M} \sum_{m \in \mathcal{M}} \boldsymbol{\varepsilon}_m^k \right\|_2^2 \leqslant \frac{\gamma}{M^2} \left\| \sum_{m \in \mathcal{M}_c^k} \boldsymbol{\varepsilon}_m^k \right\|_2^2 , \tag{29}$$

where K denotes the termination time, and $\boldsymbol{\varepsilon}_m^k = \nabla f_m(\boldsymbol{\theta}^k) - \left( \boldsymbol{q}_m^{k-1} + \Delta \boldsymbol{q}_m^k \right)$. This lemma is easy to verify when $\mathcal{M}_c^k \neq \varnothing$, a bounded variable (here is $\boldsymbol{\varepsilon}^k$) will always be bounded by a part of itself ($\frac{1}{M} \sum_{m \in \mathcal{M}_c^k} \boldsymbol{\varepsilon}_m^k$) multiplied by a real number ($\gamma$). Note that there is another nontrivial scenario that $\mathcal{M}_c^k \neq \varnothing$ but $\boldsymbol{\varepsilon}_m^k = 0$ for all $m \in \mathcal{M}_c^k$, which implies that $\gamma = 0$ or not exists and conflicts with our assumption. However, this situation only happens when all entries of $\boldsymbol{\varepsilon}_m^k = 0$, i.e., $[\nabla f_m(\boldsymbol{\theta}^k)]_i = [\boldsymbol{q}_m^{k-1}]_i$ for all $0 \leqslant i \leqslant d$.

**Lemma B.1.** *The summation of quantized gradient innovation and quantization error is bounded by the global model difference:*

$$\left\| \frac{1}{M} \sum_{m \in \mathcal{M}_c^k} \Delta \boldsymbol{q}_m^k \right\|_2^2 + \|\boldsymbol{\varepsilon}^k\|_2^2 \leqslant \frac{\beta \gamma}{\alpha^2} \left\| \boldsymbol{\theta}^k - \boldsymbol{\theta}^{k-1} \right\|_2^2 , \tag{30}$$

*Proof.*

$$
\begin{aligned}
&\left\| \frac{1}{M} \sum_{m \in \mathcal{M}_c^k} \Delta \boldsymbol{q}_m^k \right\|_2^2 + \left\| \varepsilon^k \right\|_2^2 \\
&\overset{(a)}{\leqslant} \left\| \frac{1}{M} \sum_{m \in \mathcal{M}_c^k} \Delta \boldsymbol{q}_m^k \right\|_2^2 + \gamma \left\| \frac{1}{M} \sum_{m \in \mathcal{M}_c^k} \varepsilon_m^k \right\|_2^2 \\
&\overset{(25)}{\leqslant} |\mathcal{M}_c^k| \sum_{m \in \mathcal{M}_c^k} \left\| \frac{1}{M} \Delta \boldsymbol{q}_m^k \right\|_2^2 + \gamma |\mathcal{M}_c^k| \sum_{m \in \mathcal{M}_c^k} \left\| \frac{1}{M} \varepsilon_m^k \right\|_2^2 \\
&= \frac{|\mathcal{M}_c^k|}{M^2} \sum_{m \in \mathcal{M}_c^k} \left( \left\| \Delta \boldsymbol{q}_m^k \right\|_2^2 + \gamma \left\| \varepsilon_m^k \right\|_2^2 \right) \\
&\overset{(b)}{\leqslant} \frac{|\mathcal{M}_c^k|}{M^2} \sum_{m \in \mathcal{M}_c^k} \left( \gamma \left\| \Delta \boldsymbol{q}_m^k \right\|_2^2 + \gamma \left\| \varepsilon_m^k \right\|_2^2 \right) \\
&\overset{(c)}{\leqslant} \frac{\beta \gamma |\mathcal{M}_c^k|^2}{\alpha^2 M^2} \left\| \boldsymbol{\theta}^k - \boldsymbol{\theta}^{k-1} \right\|_2^2 \\
&\leqslant \frac{\beta \gamma}{\alpha^2} \left\| \boldsymbol{\theta}^k - \boldsymbol{\theta}^{k-1} \right\|_2^2,
\end{aligned}
\tag{31}
$$

where (a) follows Assumption B.1, (b) follows $\gamma$ is larger than 1 by definition, and (c) utilizes our novel trigger condition (12). □

**Lemma B.2**. From Definition 3.1, we can derive that the relationship between **quantized** gradient innovation $\Delta \boldsymbol{q}_m^k$ and its quantization representation $\boldsymbol{\psi}_m^k$ which applies $b_m^k$ bits for each dimension:

$$
\Delta \boldsymbol{q}_m^k = 2 \tau_m^k R_m^k \boldsymbol{\psi}_m^k - R_m^k \mathbf{1},
\tag{32}
$$

where $\mathbf{1} \in \mathbb{R}^d$ denotes a vector filled with scalar value 1.

**Remark:** We can utilize (32) to calculate the quantized gradient innovation in the experimental implementation.

## C  MISSING PROOF OF LEMMA 3.1 AND THE DERIVATION OF $b_m^k$

With lazy aggregation, the actual aggregated model at epoch $k$ is:

$$
\boldsymbol{\theta}^{k+1} = \boldsymbol{\theta}^k - \frac{\alpha}{M} \sum_{m \in \mathcal{M}^k} \left( \boldsymbol{q}_m^{k-1} + \Delta \boldsymbol{q}_m^k \right) - \frac{\alpha}{M} \sum_{m \in \mathcal{M}_c^k} \boldsymbol{q}_m^{k-1}.
\tag{33}
$$

Suppose $\Delta_m^k$ denotes the **quantization loss** of device $m$ at epoch $k$ and $\boldsymbol{\psi}_m^k$ denotes the quantization representation of local gradient innovation as in Definition 3.1, i.e.,

$$
\Delta_m^k = \boldsymbol{\psi}_m^k - \frac{\nabla f_m \left( \boldsymbol{\theta}^k \right) - \boldsymbol{q}_m^{k-1} + R_m^k \mathbf{1}}{2 \tau_m^k R_m^k} - \frac{1}{2} \mathbf{1}
\tag{34}
$$

With (7), (33), and (34), the model deviation $\|\tilde{\boldsymbol{\theta}}^k - \boldsymbol{\theta}^k\|_2^2$ caused by skipping gradients can be written as:

$$
\begin{aligned}
\left\|\tilde{\boldsymbol{\theta}}^k - \boldsymbol{\theta}^k\right\|_2^2 &= \left\|\frac{\alpha}{M} \sum_{m \in \mathcal{M}_c^k} \Delta \boldsymbol{q}_m^k\right\|_2^2 \\
&= \left\|\frac{\alpha}{M} \sum_{m \in \mathcal{M}_c^k} \left(2\tau_m^k R_m^k \boldsymbol{\psi}_m^k - R_m^k \mathbf{1}\right)\right\|_2^2 \\
&\overset{(25)}{\leqslant} \frac{\alpha^2 |\mathcal{M}_c^k|}{M^2} \sum_{m \in \mathcal{M}_c^k} \left\|2\tau_m^k R_m^k \boldsymbol{\psi}_m^k - R_m^k \mathbf{1}\right\|_2^2 \\
&\overset{(34)}{\leqslant} \frac{\alpha^2 |\mathcal{M}_c^k|}{M^2} \sum_{m \in \mathcal{M}_c^k} \left(\left\|\nabla f_m(\boldsymbol{\theta}^k) - \boldsymbol{q}_m^{k-1} + R_m^k \mathbf{1} + \tau_m^k R_m^k \mathbf{1} + \Delta_m^k - R_m^k \mathbf{1}\right\|_2^2\right) \\
&\overset{(34)}{\leqslant} \frac{2\alpha^2 |\mathcal{M}_c^k|}{M^2} \sum_{m \in \mathcal{M}_c^k} \left(\left\|\nabla f_m(\boldsymbol{\theta}^k) - \boldsymbol{q}_m^{k-1} + \tau_m^k R_m^k \mathbf{1}\right\|_2^2 + \left\|\Delta_m^k\right\|_2^2\right) \\
&\overset{(a)}{\leqslant} \frac{2\alpha^2 |\mathcal{M}_c^k|}{M^2} \sum_{m \in \mathcal{M}_c^k} \left(\left\|\nabla f_m(\boldsymbol{\theta}^k) - \boldsymbol{q}_m^{k-1} + \tau_m^k R_m^k \mathbf{1}\right\|_2^2 + d\right) \\
&\overset{(28)}{\leqslant} \frac{2\alpha^2 |\mathcal{M}_c^k|}{M^2} \sum_{m \in \mathcal{M}_c^k} \left(\left(\left\|\nabla f_m(\boldsymbol{\theta}^k) - \boldsymbol{q}_m^{k-1}\right\|_2 + \left\|\tau_m^k R_m^k \mathbf{1}\right\|_2\right)^2 + d\right) \\
&= \frac{2\alpha^2 |\mathcal{M}_c^k|}{M^2} \sum_{m \in \mathcal{M}_c^k} \left(\left(\left\|\nabla f_m(\boldsymbol{\theta}^k) - \boldsymbol{q}_m^{k-1}\right\|_2 - \left\|\tau_m^k R_m^k \mathbf{1}\right\|_2 + 2\left\|\tau_m^k R_m^k \mathbf{1}\right\|_2\right)^2 + d\right) \\
&\leqslant \frac{4\alpha^2 |\mathcal{M}_c^k|}{M^2} \sum_{m \in \mathcal{M}_c^k} \left(\left(\left\|\nabla f_m(\boldsymbol{\theta}^k) - \boldsymbol{q}_m^{k-1}\right\|_2 - \left\|\tau_m^k R_m^k \mathbf{1}\right\|_2\right)^2 + 4\left\|\tau_m^k R_m^k \mathbf{1}\right\|_2^2 + \frac{d}{2}\right) \\
&\overset{(b)}{\leqslant} \frac{4\alpha^2 |\mathcal{M}_c^k|}{M^2} \sum_{m \in \mathcal{M}_c^k} \left(\left(\left\|\nabla f_m(\boldsymbol{\theta}^k) - \boldsymbol{q}_m^{k-1}\right\|_2 - \left\|\tau_m^k R_m^k \mathbf{1}\right\|_2\right)^2 + 4(R_m^k)^2 d + \frac{d}{2}\right),
\end{aligned}
$$
(35)

where $\mathbf{1} \in \mathbb{R}^d$ denotes the vector filled with scalar value 1, (a) $\Delta_m^k \in (-1, 0]$, (b) $R_m^k \geqslant \tau_m^k R_m^k \geqslant 0$.

Since $R_m^k$ is independent of $\tau_m^k$, we can formulate an optimization problem about $\tau_m^k$ for device $m$ at communication round $k$ as follows:

$$
\min_{0 < \tau_m^k \leqslant 1} \quad \left(\left\|\nabla f_m(\boldsymbol{\theta}^k) - \boldsymbol{q}_m^{k-1}\right\|_2 - \left\|\tau_m^k R_m^k \mathbf{1}\right\|_2\right)^2
$$
(36)

Therefore, the optimal solution of $\tau_m^k$ in (36) is

$$
(\tau_m^k)^* = \frac{\left\|\nabla f_m(\boldsymbol{\theta}^k) - \boldsymbol{q}_m^{k-1}\right\|_2}{R_m^k \sqrt{d}}.
$$
(37)

Then, the optimal adaptive quantization level $(b_m^k)^*$ is equal to

$$
\begin{aligned}
(b_m^k)^* &= \left\lfloor \log_2\left(\frac{1}{(\tau_m^k)^*} + 1\right) \right\rfloor \\
&= \left\lfloor \log_2\left(\frac{R_m^k \sqrt{d}}{\left\|\nabla f_m(\boldsymbol{\theta}^k) - \boldsymbol{q}_m^{k-1}\right\|_2} + 1\right) \right\rfloor
\end{aligned}
$$
(38)

Notice that $(b_m^k)^* \geqslant 1$ is always true since $(\tau_m^k)^* \leqslant 1$

# D  Missing Proof of Lemma 4.1, Theorem 4.1 and Corollary 4.1.

*Proof.* Suppose Assumptions 4.1, 4.2, and 4.3 are satisfied and $M_c^k \neq \varnothing$. For the simplicity of the convergence proof, we assume $\Phi^k = \frac{1}{M} \sum_{m \in \mathcal{M}_c^k} \Delta \boldsymbol{q}_m^k$. First, we prove Lemma 4.1.

$$
\begin{aligned}
&f(\boldsymbol{\theta}^{k+1}) - f(\boldsymbol{\theta}^k) \\
&\leqslant \left\langle \nabla f(\boldsymbol{\theta}^k), \boldsymbol{\theta}^{k+1} - \boldsymbol{\theta}^k \right\rangle + \frac{L}{2} \left\| \boldsymbol{\theta}^{k+1} - \boldsymbol{\theta}^k \right\|_2^2 \\
&= \left\langle \nabla f(\boldsymbol{\theta}^k), -\alpha \left( \nabla f(\boldsymbol{\theta}^k) - \varepsilon^k - \Phi^k \right) \right\rangle + \frac{L}{2} \left\| \boldsymbol{\theta}^{k+1} - \boldsymbol{\theta}^k \right\|_2^2 \\
&= -\alpha \left\| \nabla f(\boldsymbol{\theta}^k) \right\|_2^2 + \alpha \left\langle \nabla f(\boldsymbol{\theta}^k), \varepsilon^k + \Phi^k \right\rangle + \frac{L}{2} \left\| \boldsymbol{\theta}^{k+1} - \boldsymbol{\theta}^k \right\|_2^2 \\
&\overset{(26)}{=} -\alpha \left\| \nabla f(\boldsymbol{\theta}^k) \right\|_2^2 + \frac{\alpha}{2} \left( \left\| \nabla f(\boldsymbol{\theta}^k) \right\|_2^2 + \left\| \varepsilon^k + \Phi^k \right\|_2^2 - \frac{1}{\alpha^2} \left\| \boldsymbol{\theta}^{k+1} - \boldsymbol{\theta}^k \right\|_2^2 \right) + \frac{L}{2} \left\| \boldsymbol{\theta}^{k+1} - \boldsymbol{\theta}^k \right\|_2^2 \\
&\leqslant -\frac{\alpha}{2} \left\| \nabla f(\boldsymbol{\theta}^k) \right\|_2^2 + \frac{\alpha}{2} \left\| \varepsilon^k + \Phi^k \right\|_2^2 + \left( \frac{L}{2} - \frac{1}{2\alpha} \right) \left\| \boldsymbol{\theta}^{k+1} - \boldsymbol{\theta}^k \right\|_2^2 \\
&\overset{(25)}{\leqslant} -\frac{\alpha}{2} \left\| \nabla f(\boldsymbol{\theta}^k) \right\|_2^2 + \alpha \left\| \varepsilon^k \right\|_2^2 + \alpha \left\| \Phi^k \right\|_2^2 + \left( \frac{L}{2} - \frac{1}{2\alpha} \right) \left\| \boldsymbol{\theta}^{k+1} - \boldsymbol{\theta}^k \right\|_2^2 .
\end{aligned}
\tag{39}
$$

Hence, we have

$$
f(\boldsymbol{\theta}^{k+1}) - f(\boldsymbol{\theta}^k) \overset{(30)}{\leqslant} -\frac{\alpha}{2} \left\| \nabla f(\boldsymbol{\theta}^k) \right\|_2^2 + \left( \frac{L}{2} - \frac{1}{2\alpha} \right) \left\| \boldsymbol{\theta}^{k+1} - \boldsymbol{\theta}^k \right\|_2^2 + \frac{\beta\gamma}{\alpha} \left\| \boldsymbol{\theta}^k - \boldsymbol{\theta}^{k-1} \right\|_2^2 , \tag{40}
$$

which gives us Theorem 4.1. Sum it up for $k = 1, 2, \cdots, K$, we have

$$
\begin{aligned}
f(\boldsymbol{\theta}^{K+1}) - f(\boldsymbol{\theta}^1) \leqslant{}& -\frac{\alpha}{2} \sum_{k=1}^{K} \left\| \nabla f(\boldsymbol{\theta}^k) \right\|_2^2 + \left( \frac{L}{2} - \frac{1}{2\alpha} \right) \left\| \boldsymbol{\theta}^{K+1} - \boldsymbol{\theta}^K \right\|_2^2 \\
&+ \sum_{k=1}^{K-1} \left( \frac{L}{2} - \frac{1}{2\alpha} + \frac{\beta\gamma}{\alpha} \right) \left\| \boldsymbol{\theta}^{k+1} - \boldsymbol{\theta}^k \right\|_2^2 + \frac{\beta\gamma}{\alpha} \left\| \boldsymbol{\theta}^1 - \boldsymbol{\theta}^0 \right\|_2^2 .
\end{aligned}
\tag{41}
$$

Notice that inequality (41) holds for both $M_c^k \neq \varnothing$ and $M_c^k = \varnothing$. Therefore, for $\left( \frac{L}{2} - \frac{1}{2\alpha} + \frac{\beta\gamma}{\alpha} \right) \leqslant 0$ and all hyperparameters are chosen properly, considering the minimum of $\| \nabla f(\boldsymbol{\theta}^k) \|_2^2$

$$
\begin{aligned}
\min_{k=1,2,\cdots,K} \left\| \nabla f(\boldsymbol{\theta}^k) \right\|_2^2 \leqslant{}& \frac{1}{K} \sum_{k=1}^{K} \left\| \nabla f(\boldsymbol{\theta}^k) \right\|_2^2 \\
&\overset{(41)}{\leqslant} \frac{2}{\alpha K} \left( f(\boldsymbol{\theta}^1) - f(\boldsymbol{\theta}^K) + \frac{\beta\gamma}{\alpha} \left\| \boldsymbol{\theta}^1 - \boldsymbol{\theta}^0 \right\|_2^2 \right) .
\end{aligned}
\tag{42}
$$

For $\left( \frac{L}{2} - \frac{1}{2\alpha} + \frac{\beta\gamma}{\alpha} \right) \leqslant 0$ and all hyperparameters are chosen properly, we have that

$$
\min_{k=1,2,\cdots,K} \left\| \nabla f(\boldsymbol{\theta}^k) \right\|_2^2 \leqslant \frac{2}{\alpha K} \left( f(\boldsymbol{\theta}^1) - f(\boldsymbol{\theta}^*) + \frac{\beta\gamma}{\alpha} \left\| \boldsymbol{\theta}^1 - \boldsymbol{\theta}^0 \right\|_2^2 \right) \leqslant \epsilon^2 , \tag{43}
$$

which demonstrates AQUILA requires $K = \mathcal{O}\left( \frac{2\omega_1}{\alpha\epsilon^2} \right)$ communication round with $\omega_1 = f(\boldsymbol{\theta}^1) - f(\boldsymbol{\theta}^*) + \frac{\beta\gamma}{\alpha} \left\| \boldsymbol{\theta}^1 - \boldsymbol{\theta}^0 \right\|_2^2$ to achieve $\min_{k=1,2,\cdots,K} \left\| \nabla f(\boldsymbol{\theta}^k) \right\|_2^2 \leqslant \epsilon^2$. $\qquad \square$

# E    MISSING PROOF OF COROLLARY 4.1 WHEN $M_c^k = \varnothing$.

*Proof.* Since the skipping subset of devices are the empty set, from (5), we have

$$
\begin{aligned}
\boldsymbol{\theta}^{k+1} - \boldsymbol{\theta}^k &= -\frac{\alpha}{M} \sum_{m \in \mathcal{M}^k} \left( \boldsymbol{q}_m^{k-1} + \Delta \boldsymbol{q}_m^k \right) - \frac{\alpha}{M} \sum_{m \in \mathcal{M}_c^k} \boldsymbol{q}_m^{k-1} \\
&= -\frac{\alpha}{M} \sum_{m \in \mathcal{M}} \left( \boldsymbol{q}_m^{k-1} + \Delta \boldsymbol{q}_m^k \right) \\
&\overset{(11)}{=} -\frac{\alpha}{M} \sum_{m \in \mathcal{M}} \left( \nabla f_m(\boldsymbol{\theta}^k) - \varepsilon_m^k \right) \\
&= -\alpha \left( \nabla f(\boldsymbol{\theta}^k) - \varepsilon^k \right).
\end{aligned}
\tag{44}
$$

From (14) we have:

$f(\boldsymbol{\theta}^{k+1}) - f(\boldsymbol{\theta}^k)$

$$
\begin{aligned}
&\leqslant -\frac{\alpha}{2} \left\| \nabla f(\boldsymbol{\theta}^k) \right\|_2^2 + \alpha \left\| \frac{1}{M} \sum_{m \in \mathcal{M}_c^k} \Delta \boldsymbol{q}_m^k \right\|_2^2 + \left( \frac{L}{2} - \frac{1}{2\alpha} \right) \left\| \boldsymbol{\theta}^{k+1} - \boldsymbol{\theta}^k \right\|_2^2 + \alpha \left\| \varepsilon^k \right\|_2^2 \\
&\leqslant -\frac{\alpha}{2} \left\| \nabla f(\boldsymbol{\theta}^k) \right\|_2^2 + \left( \frac{L}{2} - \frac{1}{2\alpha} \right) \left\| \boldsymbol{\theta}^{k+1} - \boldsymbol{\theta}^k \right\|_2^2 + \alpha \left\| \varepsilon^k \right\|_2^2 \\
&\overset{(27)}{\leqslant} -\frac{\alpha}{2} \left\| \nabla f(\boldsymbol{\theta}^k) \right\|_2^2 + \alpha^2 \left( \frac{L}{2} - \frac{1}{2\alpha} \right) \left( (1+p) \left\| \nabla f(\boldsymbol{\theta}^k) \right\|_2^2 + (1+p^{-1}) \left\| \varepsilon^k \right\|_2^2 \right) + \alpha \left\| \varepsilon^k \right\|_2^2 \\
&= -\frac{\alpha}{2} \left\| \nabla f(\boldsymbol{\theta}^k) \right\|_2^2 + \frac{1}{2} \left( \alpha^2 L - \alpha \right) (1+p) \left\| \nabla f(\boldsymbol{\theta}^k) \right\|_2^2 + \frac{1}{2} \left( \alpha^2 L - \alpha \right) (1+p^{-1}) \left\| \varepsilon^k \right\|_2^2 + \alpha \left\| \varepsilon^k \right\|_2^2 \\
&= \frac{\alpha}{2} \left( (\alpha L - 1)(1+p) - 1 \right) \left\| \nabla f(\boldsymbol{\theta}^k) \right\|_2^2 + \frac{\alpha}{2} \left( (\alpha L - 1)(1+p^{-1}) + 2 \right) \left\| \varepsilon^k \right\|_2^2.
\end{aligned}
\tag{45}
$$

If the factor of $\left\| \varepsilon^k \right\|_2^2$ in (45) is less than or equal to 0, that is,

$$
(\alpha L - 1)(1 + p^{-1}) + 2 \leqslant 0,
\tag{46}
$$

then the factor of $\|\nabla f(\boldsymbol{\theta}^k)\|_2^2$ will be less than $-\frac{\alpha}{2}$, which indicates that

$$
f(\boldsymbol{\theta}^{k+1}) - f(\boldsymbol{\theta}^k) \leqslant -\frac{\alpha}{2} \left\| \nabla f(\boldsymbol{\theta}^k) \right\|_2^2.
\tag{47}
$$

Note that it is not difficult to demonstrate that (46) and $\frac{L}{2} - \frac{1}{2\alpha} + \frac{\beta\gamma}{\alpha} \leqslant 0$ can actually be satisfied at the same time. For instance, we can set $p = 0.1, \alpha = 0.1, \beta = 0.25, \gamma = 2, L = 2.5$ that satisfies both of them. □

# F    MISSING PROOF OF THEOREM 4.2.

*Proof.* Based on the intermediate result (40) of Theorem 4.1 and Assumption 4.4 ($\mu$−PŁ condition), we have

$$
\begin{aligned}
f(\boldsymbol{\theta}^{k+1}) - f(\boldsymbol{\theta}^k) &\leqslant -\frac{\alpha}{2} \left\| \nabla f(\boldsymbol{\theta}^k) \right\|_2^2 + \left( \frac{L}{2} - \frac{1}{2\alpha} \right) \left\| \boldsymbol{\theta}^{k+1} - \boldsymbol{\theta}^k \right\|_2^2 + \frac{\beta\gamma}{\alpha} \left\| \boldsymbol{\theta}^k - \boldsymbol{\theta}^{k-1} \right\|_2^2 \\
&\overset{(19)}{\leqslant} -\alpha\mu(f(\boldsymbol{\theta}^k) - f(\boldsymbol{\theta}^*)) + \left( \frac{L}{2} - \frac{1}{2\alpha} \right) \left\| \boldsymbol{\theta}^{k+1} - \boldsymbol{\theta}^k \right\|_2^2 + \frac{\beta\gamma}{\alpha} \left\| \boldsymbol{\theta}^k - \boldsymbol{\theta}^{k-1} \right\|_2^2,
\end{aligned}
\tag{48}
$$

which is equivalent to

$$
f(\boldsymbol{\theta}^{k+1}) - f(\boldsymbol{\theta}^*) \overset{(19)}{\leqslant} (1 - \alpha\mu)(f(\boldsymbol{\theta}^k) - f(\boldsymbol{\theta}^*)) + \left( \frac{L}{2} - \frac{1}{2\alpha} \right) \left\| \boldsymbol{\theta}^{k+1} - \boldsymbol{\theta}^k \right\|_2^2 + \frac{\beta\gamma}{\alpha} \left\| \boldsymbol{\theta}^k - \boldsymbol{\theta}^{k-1} \right\|_2^2.
\tag{49}
$$

Suppose $\frac{\beta\gamma}{\alpha} \leqslant (1-\alpha\mu)\left(\frac{1}{2\alpha}-\frac{L}{2}\right)$, we can show that

$$
\begin{aligned}
f(\boldsymbol{\theta}^{k+1}) - f(\boldsymbol{\theta}^*) + \left(\frac{1}{2\alpha}-\frac{L}{2}\right)&\left\|\boldsymbol{\theta}^{k+1}-\boldsymbol{\theta}^k\right\|_2^2 \\
&\leqslant (1-\alpha\mu)\left(f(\boldsymbol{\theta}^k)-f(\boldsymbol{\theta}^*)+\left(\frac{1}{2\alpha}-\frac{L}{2}\right)\left\|\boldsymbol{\theta}^k-\boldsymbol{\theta}^{k-1}\right\|_2^2\right).
\end{aligned}
\tag{50}
$$

Therefore, after multiply $k = 1, 2, \cdots, K$, we have

$$
\begin{aligned}
f(\boldsymbol{\theta}^{K+1}) - f(\boldsymbol{\theta}^*) + \left(\frac{1}{2\alpha}-\frac{L}{2}\right)&\left\|\boldsymbol{\theta}^{K+1}-\boldsymbol{\theta}^K\right\|_2^2 \\
&\leqslant (1-\alpha\mu)^K\left(f(\boldsymbol{\theta}^1)-f(\boldsymbol{\theta}^*)+\left(\frac{1}{2\alpha}-\frac{L}{2}\right)\left\|\boldsymbol{\theta}^1-\boldsymbol{\theta}^0\right\|_2^2\right) \leqslant \epsilon,
\end{aligned}
\tag{51}
$$

which demonstrates AQUILA requires $K = \mathcal{O}\left(-\frac{1}{\log(1-\alpha\mu)}\log\frac{\omega_1}{\epsilon}\right)$ communication round with $\omega_1 = f(\boldsymbol{\theta}^1)-f(\boldsymbol{\theta}^*)+\left(\frac{1}{2\alpha}-\frac{L}{2}\right)\|\boldsymbol{\theta}^1-\boldsymbol{\theta}^0\|_2^2$ to achieve $f(\boldsymbol{\theta}^{K+1})-f(\boldsymbol{\theta}^*)+\left(\frac{1}{2\alpha}-\frac{L}{2}\right)\|\boldsymbol{\theta}^{K+1}-\boldsymbol{\theta}^K\|_2^2 \leqslant \epsilon$. $\qquad\square$

