# OpenReview forum: "AQUILA: Communication Efficient Federated Learning with Adaptive Quantization of Lazily-Aggregated Gradients"
_ICLR.cc/2023/Conference — Submitted to ICLR 2023_

### Official Review · Reviewer_NDFY · 2022-10-24

**Confidence:** 4
**Clarity, Quality, Novelty And Reproducibility:** Please see above.
**Correctness:** 3
**Technical Novelty And Significance:** 2
**Empirical Novelty And Significance:** 2
**Recommendation:** 3

**Strength And Weaknesses:**

Pros:
1. The topic of federated learning (FL) is important and interesting to NeurIPS community.
2. The paper provides both theoretical and empirical results.
3. The writing and presentation, though can still be improved, is in general clear.

Cons:
1. (i) The motivation is not very strong. While alternating the number of bits $b$ across training rounds might be effective, it is not guaranteed to always bring benefits (reduced communication). Specifically, in formula (10) when choosing $b^*$, it seems that the number of bits largely relies on the largest entry of the vector ($l_\infty$ norm). What if the vector only has a few 'outliers'? This strategy would use a large $b$ to compromise those outliers. A more practical and possibly better strategy might be sending those large coordinates in full-precision, and quantize others using low bits. This may use much less communication than the proposed strategy. In my understanding, (10) basically says that if the largest magnitude of the vector is big, we use more bits. This does not look very exciting and promising to me.

(ii) From Figure 2 and Figure 3, we see that the communication across training rounds of AQUILA is almost constant (a flat line). Then why not simply using a fixed number of bits? I think a figure with the evolution of the $b^*$ for the clients might help.

(iii) The communication criteria (11) looks more like a byproduct for the convenience of proof (e.g., Lemma A.1). Could you please provide more intuition on it and its difference with (6)? Besides, I have two questions regarding (11). First, how do you know $\theta^{k+1}$ at time $k$? Second, are both $\gamma$ and $\beta$ free hyperparameters? If so, the algorithm becomes much harder to tune which is also a potential drawback.

2. The convergence analysis considers full gradient without noise, which is less practical since people use SGD in most applications. On page 6, the authors wrote 'we could still prove the correctness of Corollary 4.1 in this special condition without any extra assumptions'. Then why is Assumption 4.3 needed?

3. Indeed, Assumption 4.3 seems a little bit strange. In (13), is the $\gamma$ the same as that in (11)? Also, you can assume a very large $\gamma$ for this condition to hold. However, I do not see this $\gamma$ appearing in later theoretical results. So how does this assumption affect the analysis?

4. The experiments uses $M=24$ clients which is rather small for FL. Also, there is no ablation study on the impact of $\gamma$ and $\beta$. Thus we do not know how robust this algorithm is to different choices of the hyperparameters. I think the authors should also compare with strategies with fixed-bit quantization like the popular QSGD method. This could help justify the importance of adaptive quantization and strengthen the motivation. Currently, given the above concerns, I think the motivation is not very strong.

**Summary Of The Paper:**

The paper proposes a new adaptive quantization strategy (and aggregation criteria), AQUILA, by adaptively determining the quantization bits per round per client in lazily aggreagated federated learning. The quantization level $b^*$ is chosen based on the $l_\infty$ and $l_2$ norm of the difference between the current gradient and the previous quantized gradient. Convergence rates in non-convex and PL condition are provided. Experiments are conducted to show the advantages of the proposed method.

**Summary Of The Review:**

I feel that the motivation of this paper is not very strong. I have some concerns about the design of the new adaptive quantization strategy. The theoretical analysis needs some seemingly meaningless assumption and does not consider stochastic gradients which is less practical. The experiments lacks ablation study and comparison with important methods. Thus, I think the paper is below the high bar of ICLR in its current version.

---

> ### Author Response · Authors · 2022-11-18
> **We thank the reviewer for their interesting comments! Please see our rebuttal below.**
>
> > Q1: We need to use more bits when a few outliers occur.
>
> Re: Thank you for your professional review and constructive suggestions. The outliers may not occur very frequently in the practical scenarios since, during the entire training rounds, the change in gradient will not be particularly large (smoothness), resulting in most of the values of $\nabla f_m(\boldsymbol{\theta}^k)-\boldsymbol{q}_m^{k-1}$ being in a relatively concentrated area. Hence, the number of outliers will not be enormous, and their value will not be too far from others. Furthermore, when the $ \ell_\infty$ norm of $\nabla f_m(\boldsymbol{\theta}^k)-\boldsymbol{q}_m^{k-1} $ (numerator) becomes larger, the $\ell_2$ norm of$\nabla f_m(\boldsymbol{\theta}^k)-\boldsymbol{q}_m^{k-1}$ (denominator) will also become larger. Additionally, the log operation in calculating $(b_m^k)^*$ also weakens the influence of the outlier and thus makes the final result not much higher than that without outliers.
>
> > Q2: The communication across training rounds of AQUILA is an almost fixed number (a flat line).
>
> Re: The solid lines represent the value after smoothing, and transparent shades of the same colors around them represent the true value. Indeed, the transmitted bits of each client are varied from all training rounds rather than a fixed number. To make it less misleading, we remove all transparent shades in the original figures and keep only AQUILA's. These removed figures can be found in Appendix A.4.
>
> > Q3: What is the difference between AQUILA’s skip rule (11) and the LAQ’s (24)?
>
> Re: Thanks for bringing up this concern. The difference between AQUILA skipping criterion and LAQ skipping criterion is as follows.
>
> First, the AQUILA threshold is easier to compute for a local client. Compared to the LAQ skipping criterion, AQUILA skipping criterion is more concise and thus requires less storage and computing power.
>
> Second, the AQUILA criterion is easier to tune because significantly fewer hyperparameters are introduced. Compared to the LAQ criterion in which  $\alpha$, $D$  and  $\\{\xi_{d^{\prime}}\\}_{d^{\prime}=1}^D$ are all manually selected, whilst only two hyperparameters $\alpha$  and $\beta$ are introduced in the AQUILA criterion.
>
> Third, with the given threshold, AQUILA has a good theoretical property. The theoretical analysis of AQUILA is easier to follow with no Lyapunov function introduced as in LAQ. And the result also shows that  AQUILA can achieve a better convergence rate under the non-convex case and the PL condition.
>
> > Q4: Theoretical analysis of stochastic gradient descent.
>
> Re: Thanks for your constructive comment. For most literature that theoretically analyzes the stochastic behaviors of the quantization strategies, such as [1, 2], their objective quantizers are usually stochastic quantizers with unbiased and bounded variance assumptions. However, our paper utilizes a biased deterministic quantizer, making it difficult to analyze its expectation under the stochastic gradient scenario (due to the round-down operation) [3, 4]. This is a significant challenge, and we will consider it as our future work.
>
> > Q5: Why not $\gamma$ appears in theoretical results?
>
> Re:  Although $\gamma $  does not directly appear in the convergence rate, we still need it to derive the convergence rate based on the conditions: $\frac{L}{2}-\frac{1}{2 \alpha}+\frac{\beta\gamma}{\alpha} \leqslant 0$ and  $\frac{\beta\gamma}{\alpha} \leqslant (1-\alpha\mu)\left(\frac{1}{2\alpha} - \frac{L}{2}\right)$ in Theorem 4.1 and 4.2, respectively.
>
> Considering the coherence of the reader, we have moved the original Assumption 4.3 to the Appendix, Assumption B.1, to make the logical structure of the whole article more general and smooth.
>
> > Q6: 24 clients are insufficient, needing more clients in the FL setting; ablation study on hyperparameter; fixed-bit quantization like QSGD.
>
> Re: We thank the reviewer for the valuable suggestion. In the revision, we have conducted the evaluation of ***100*** devices for the CIFAR-10 and CIFAR-100 datasets and ***80*** devices for the WikiText-2 dataset on 4 NVIDIA RTX A6000 GPUs (total 192G VRAM). We want to emphasize that although many existing works seem to use a large number of clients, in fact, only a small part is chosen in each training round.
>
> Furthermore, we have added a new part on studying the effect of the hyperparameter  $\beta$ in Section 5.4 of the revision, and suitable selections of $\beta$ value are provided.
>
> Finally, we have added the implementation of QSGD for all FL settings in our evaluation part, and the results demonstrate that AQUILA still has the best performance among all the compared algorithms.

---

> > ### Author Response · Authors · 2022-11-18
> > **Q7 and References**
> >
> > > Q7: Require knowledge of $\boldsymbol{\theta}^{k+1}$ in round k.
> >
> >
> > Re: Thank you for pointing out this problem. This is a symbol misuse, and we have changed it to, for $k \geqslant 1 $,
> >
> > $$
> > \left\\| \Delta \boldsymbol{q}_{m}^{k}\right\\|_2^2+ \left\\|\boldsymbol{\varepsilon}_m^{k}\right\\|_2^2 \leqslant \frac{\beta}{ \alpha^2}\left\\| \boldsymbol{\theta}^{k}-\boldsymbol{\theta}^{k-1} \right\\|_2^2, \forall m \in \mathcal{M}_c^k.
> > $$
> >
> > And it is emphasized in the paper that the device selection strategy works in the k-th round.
> >
> > > **References**
> >
> > [1] Gorbunov E, Burlachenko K P, Li Z, et al. MARINA: Faster non-convex distributed learning with compression[C]//International Conference on Machine Learning. PMLR, 2021: 3788-3798.MLA
> >
> > [2] Ghadikolaei H S, Stich S, Jaggi M. LENA: Communication-efficient distributed learning with self-triggered gradient uploads[C]//International Conference on Artificial Intelligence and Statistics. PMLR, 2021: 3943-3951.MLA
> >
> > [3] Li H, De S, Xu Z, et al. Training quantized nets: A deeper understanding[J]. Advances in Neural Information Processing Systems, 2017, 30.MLA
> >
> > [4] Zhu S, Chen B. Quantized consensus by the ADMM: Probabilistic versus deterministic quantizers[J]. IEEE Transactions on Signal Processing, 2015, 64(7): 1700-1713.

---

### Official Review · Reviewer_ogm4 · 2022-10-25

**Confidence:** 2
**Correctness:** 4
**Technical Novelty And Significance:** 2
**Empirical Novelty And Significance:** 2
**Recommendation:** 6

**Clarity, Quality, Novelty And Reproducibility:**

The paper is well written. The authors share experimental details and the code to reproduce the results.

**Strength And Weaknesses:**

The paper is well-written. Overall, I enjoyed reading the paper. Even though the main ideas and the approaches in the proofs are mainly borrowed from existing works, I think the authors did a good job in combining the methods.

- When averaging the local objective functions in (1), why isn't the local function weighted by the number of local samples $n_m$?

- Page 4, first line: quantification -> quantization.

**Summary Of The Paper:**

The paper proposes a communication-efficient federated learning strategy, AQUILA, that combines lazily-aggregated quantization (LAQ) and adaptive quantization (AdaQuantFL). This strategy balances the communication frequency for each parameter and the quantization level adaptively. Experimental results show that AQUILA outperforms the baselines (including LAQ and AdaQuantFL individually) on CIFAR-10, CIFAR-100, and WikiText datasets.

**Summary Of The Review:**

Please see my comments above.

---

> ### Author Response · Authors · 2022-11-18
> **We thank the reviewer for their helpful comments! Please see our rebuttal below.**
>
> > Q1: Why isn't the local function weighted by the number of local samples $n_m$?
>
> Re: Thanks for raising this question. We consider the balanced data averaging setting during theoretical analysis, just like most works do (e.g. [1, 2]). However, in our evaluation part, we have evaluated the heterogeneous data distribution (i.e., different sample sizes).
>
> > Q2: typo: quantification -> quantization.
>
> Re: Thanks for your careful reading. We have already corrected it.
>
> **References**
>
> [1] Gorbunov E, Burlachenko K P, Li Z, et al. MARINA: Faster non-convex distributed learning with compression[C]//International Conference on Machine Learning. PMLR, 2021: 3788-3798.MLA
>
> [2] Richtárik P, Sokolov I, Gasanov E, et al. 3PC: Three point compressors for communication-efficient distributed training and a better theory for lazy aggregation[C]//International Conference on Machine Learning. PMLR, 2022: 18596-18648.MLA

---

### Official Review · Reviewer_qabV · 2022-10-31

**Confidence:** 3
**Correctness:** 4
**Technical Novelty And Significance:** 3
**Empirical Novelty And Significance:** 3
**Recommendation:** 6

**Clarity, Quality, Novelty And Reproducibility:**

The overall presentation is clear but the explanation of the algorithm is quite hard to follow as some background information is missing and the explanation is spread over Section 3 of the main paper and Appendices A1, B and C. I have the following suggestion for addressing this issue:

a) Consider putting Algorithm 1 in the main paper. It is much easier to follow the explanation in Section 3 after reading Algorithm 1. If space is an issue, Figure 1 can be moved to the Appendix in my opinion.

b) Please formally define gradient innovation, with the relevant notation, instead of just explaining it in one line in the introduction. The term is used several times in Section 3, and I don't think it is a commonly known term.

b) Please provide some background or intuition behind (8) and (9). While these appear to be obtained from the quantizer in LAQ, I believe there should at least be some intuition provided for these expressions for the benefit of readers unfamiliar with LAQ.

c) Please provide some justification for (10) in the main paper by suggesting how it follows from (8) and (9). Currently it seems to appear out of nowhere unless one reads the derivation in Appendix C.

d) There is also no motivation for (11) and I still do not understand why it is a suitable criterion for device groups being skipped. Specifically, it seems to require knowledge of $\theta^{k+1}$ at all worker nodes in round $k$ but since the nodes do not have access to each other's gradients I do not see how they can calculate the threshold in (11). Please clarify.


**Strength And Weaknesses:**

Strengths:

a) The main strength of the paper is that it combines adaptive quantization and lazy aggregation of gradients in a principled manner. Specifically, the quantization level at each round at a given node is clearly connected to the amount of new information (gradient innovation) being provided by the node instead of just depending on the loss value.

b) The principled approach is then backed by analysis and experiments to show that it indeed reduces the amount of data being communicated while providing the same training performance in a variety of settings.

Weaknesses:

a) The main weakness according to me is that the explanation of the algorithm is currently not very clear. I have provided some comments/suggestion on improving this in the following section.

b) While the main selling point for AQUILA is the reduction in the amount of data being communicated, the current theoretical results only show that the training converges at the same rate as standard GD. Any analytical result characterizing the expected reduction in amount of data being communicated would significantly strengthen this work. Alternately can you provide some explanation for why such an analysis is difficult?

c) Likewise, from Figure 3 (d-f) it appears that the variance in the amount of data being communicated is also significantly lower in AQUILA which is a point in its favour, especially if the network cannot handle large fluctuations in the amount of communication. Can you provide some analysis/intuition for why that may be the case?

**Summary Of The Paper:**

The paper proposes a new communication efficient federated learning approach that combines adaptive quantization with lazy aggregation of gradients at each round to reduce the amount of data communicated as compared to prior work while providing comparable reduction in training loss. The approach is validated through theoretical analysis and experiments in both homogenous and heterogeneous federated learning settings.

**Summary Of The Review:**

The paper proposes a novel principled approach for communication efficient federated learning approach that combines adaptive quantization with lazy aggregation of gradients and is backed by both theoretical analysis and experimental evaluation. However, some points need to be explained more clearly and some background information needs to be provided to make it fully accessible to readers.

---

> ### Author Response · Authors · 2022-11-18
> **We thank the reviewer for their constructive comments! Please see our rebuttal below.**
>
> > Q1: Move Fig 1 to the Appendix and Algorithm 1 to the main part.
>
> Re: Thanks for your valuable suggestion. We have already exchanged the Algorithm 1 to the main part of the paper.
>
> > Q2: Any analytical result of the reduction in the amount of data?
>
> Re: The analysis of communicated bits is difficult to perform for the following reasons.
>
> First, for different clients, the bits in different communication rounds vary a lot. And according to our bits selection method, bits are highly related to local gradients, which also fluctuate significantly. Second, lazily aggregation method skips many unimportant gradient innovations. The skipping criterion is easy to implement but hard to analyze, which makes theoretical proof even more complicated.
>
> Instead, we performed a series of experiments to support AQUILA’s reduction in communicated data in Section 5 and Appendix A.4.
>
> > Q3: Provide some intuition for why AQUILA is able to handle large fluctuations.
>
> Re: First, from the experiment results, we find that as the training progresses, the number of skipping uploading (i.e., lazy aggregation) will gradually decrease, which leads to less bit fluctuation in the later stage of AQUILA.
>
> Second, AdaQuantFL only uses loss value to calculate the quantization level, and this method will cause large fluctuations due to large changes in the loss in difficult tasks. The way AQUILA chooses  $(b^k)^*$  is more stable, and it is determined by a variety of factors. Experiments also verify this intuition.
>
> > Q4: Please formally define the gradient innovation.
>
> Re: Thank you for bringing this commitment. We have already added it to Definition 2.1.
>
> > Q5: Provide some intuition for the LAQ quantizer Eq.(6) and its difference with Eq.(32) (Lemma B.2).
>
> Re: Thanks for your commitment. The quantizer is a deterministic quantizer that, at each dimension, maps the gradient innovation to the closest point at a one-dimensional grid. The range of the grid is  $R_m^k$, and the granularity is determined by quantization level $\tau_m^k$. Each dimension of gradient innovation is mapped to an integer in {$0, 1, 2, 3, …, 2^b-1$}. More precisely, the $1/2$  ensures mapping to the closest integer instead of flooring to a smaller integer. The $R_m^k$ in the numerator ensures that the mapped integer is non-negative. As a result, when the gradient innovation is transmitted to the central server, $32$ bits are used for the range, and $b * d$ bits are used for the mapped integer. Thus, $32 + b*d$ bits are transmitted in total.
>
> The difference between (6) and (Lemma B.2) is that (6) encodes the raw gradient innovation vector to an integer vector, whilst (Lemma B.2) decodes the integer vector to a quantized gradient innovation vector. Specifically, in the training process, each client utilizes (6) to encode the gradient innovation to an integer at each dimension. Afterward, the integer vector $\boldsymbol{\psi}_m^k$  and $\tau_m^k$ are sent to a central server. After receiving them, the central server can decode the quantized gradient innovation as (Lemma B.2) states.
>
> > Q6: Provide some justification for (10) by suggesting how it follows from (8) and (9).
>
> Re: Thanks for your constructive comment. We have realized the inadequacy of the previous statement and rearranged the structure of Section 3.1. Particularly, in Section 3.1, we start with the model derivation between the fully-aggregated $\tilde{\boldsymbol{\theta}}^k$ and skipped model $\boldsymbol{\theta}^k$ and study the impact of lazy aggregation on each training round. After that, we formulate an optimization problem to minimize the upper bound of this impact, which implies the optimal criterion of selecting $(b_m^k)^*$.
>
> > Q7: Require knowledge of $\boldsymbol{\theta}^{k+1}$ in round k.
>
>
> Re: Thank you for pointing out this confusion. This is a symbol misuse, and we have changed it to
>
> $$
> \left\\| \Delta \boldsymbol{q}_{m}^{k}\right\\|_2^2+ \left\\|\boldsymbol{\varepsilon}_m^{k}\right\\|_2^2 \leqslant \frac{\beta}{ \alpha^2}\left\\| \boldsymbol{\theta}^{k}-\boldsymbol{\theta}^{k-1} \right\\|_2^2, \forall m \in \mathcal{M}_c^k.
> $$
>
> And it is emphasized in the paper that the device selection strategy works in the k-th round.

---

> > ### Comment · Reviewer_qabV · 2022-12-01
> > **Re**
> >
> > Thank you for updating the paper. The writing has significantly improved now and all the points that appeared confusing earlier have been addressed adequately in my opinion. As I had already recommended acceptance of the paper, I will keep my score.

---

### Official Review · Reviewer_6iqd · 2022-11-04

**Confidence:** 3
**Correctness:** 3
**Technical Novelty And Significance:** 2
**Empirical Novelty And Significance:** 2
**Recommendation:** 5

**Clarity, Quality, Novelty And Reproducibility:**

The paper is marginally novel compared to the existing work on quantized federated learning in the following works:

- Jun Sun, Tianyi Chen, Georgios B Giannakis, Qinmin Yang, and Zaiyue Yang. Lazily aggregated quantized gradient innovation for communication-efficient federated learning. IEEE Transactions on Pattern Analysis & Machine Intelligence, pp. 1–15, 2020.

- Tianyi Chen, Georgios B Giannakis, Tao Sun, and Wotao Yin. Lag: Lazily aggregated gradient for communication-efficient distributed learning. In Proceedings of Advances in Neural Information Processing Systems, pp. 1–25, 2018.

**Strength And Weaknesses:**

Weaknesses:

- My main concern is regarding the novelty of the proposed method and also, in improving the existing results. This paper seems to be significantly built on a prior work "Lazily Aggregated Quantized Gradient Innovation for Communication-Efficient Federated Learning". It is not clear how the performance of LAQ in the mentioned work is improved in theory. The authors only mention a comparison after Theorem 4.2: " Theorem 4.2 informs us that AQUILA can achieve a linear convergence rate as in LAG (Chen et al., 2018) and LAQ when certain assumptions are satisfied." I wonder what would be the motivation to use the proposed AQUILA method if it achieves the same performance for already existing LAG and LAQ methods.

- The paper considers a (not class) specific quantizer which limits the scope of the theoretical and simulation results.

- Assumption 4.3 states that all devices’ quantization errors are constrained by the total error of the omitted devices. The justification for this assumption is presented as follows: This assumption is easy to verify when M_c≠∅ this a bounded variable will always be bounded by a part of itself multiplied by a real number. This statement is not accurate and should be: This assumption is easy to verify when M_c≠∅ this a bounded variable will always be bounded by a non-zero part of itself multiplied by a real number. So, how can we guarantee that the total error of the omitted devices is non-zero?

**Summary Of The Paper:**

This paper proposes a framework for federated learning by adjusting the frequency of communication between agents and the server with an adaptive quantization scheme. Specifically, the authors combine two quantization schemes, namely, the adaptive quantization rule (AdaQuantFL) and lazily aggregated quantization (LAQ).

**Summary Of The Review:**

The paper falls short to highlight the main novelty compared to the previous works on the lazy aggregation of quantized gradients for federated learning as I detailed above. The theoretical results suggest no improvement compared to existing methods such as LAG and LAQ.

---

> ### Author Response · Authors · 2022-11-18
> **Thank you very much for you comments! Please see our rebuttal below.**
>
> > Q1: Novelty: the convergence comparison between AQUILA and LAQ is not clear.
>
> Re: Thanks for your constructive comment. When we mention that “AQUILA can achieve a linear convergence rate as in LAG”, we actually mean AQUILA has the same Q-(sub)linearity as LAG under (non-convex)PL condition. Indeed, there are some misleadings since only the order of the two rates is the same, but the constants inside are still different. Hence, we elaborately compared the difference in convergence rate between AQUILA and LAG in Section 4 of the revision.
>
> > Q2: The paper considers a (not class) specific quantizer.
>
> Re: Thanks for bringing up this concern. We consider the deterministic mid-tread quantizer for two reasons. First, as we said in our original version, we use a similar quantizer to LAQ for better comparison. Second, the deterministic mid-tread quantizer is a better version of a class of **deterministic rounding quantizers** with the quantization level s and a tuning factor $\kappa$:
>
> $$
> \varphi_i({\nabla f(\boldsymbol{\theta}^k)}, s, \kappa)=\left\lfloor\frac{\kappa \left|\nabla f(\boldsymbol{\theta}^k)_i\right|}{s \|\nabla f(\boldsymbol{\theta}^k)\|_p}\right\rfloor,
> $$
>
> because the mid-tread quantizer can map every entry of the gradient to its closest integer and confine the result in a certain range owing to the infinity norm.
>
> Furthermore, the derivation of AQUILA is generalizable (i.e., minimizing the upperbound of the model deviation) and can be readily transferred to other **deterministic rounding quantizers**.
>
> > Q3: How can we guarantee that the total error is non-zero in Assumption 4.3?
>
> Re: Thanks for your question. The total error is zero when $ \mathcal{M}_c^k \neq \varnothing$ implies that for all $m \in \mathcal{M}_c^k$, all entries of $\boldsymbol{\varepsilon}_m^k = 0$, i.e., $\left[\nabla f_m(\boldsymbol{\theta}^k)\right]_i = \left[\boldsymbol{q}_m^{k-1}\right]_i$
>  for all $0 \leq i \leq d$. Indeed, this condition is arduous to meet for two reasons. First, corresponding to Eq.(6), the gradient $\nabla f_m(\boldsymbol{\theta}^k)$ changes every time during the training process. Second, the gradient dimension is still rather tremendous, and the probability of each element of two random high-dimensional real vectors being equal is very low. Nevertheless, for the rigorousness of the whole paper, we still consider it an assumption (i.e., suppose there always exists a real value $\gamma$) rather than a fact.

---

### Author Response · Authors · 2022-11-18
**Primary changes in the revision and general response**

We thank all the reviewers for the helpful and constructive feedback! In the revision, we have made relatively large changes according to your suggestions. The specific changes are as follows:

- **Introduction and background**: We have deleted some confusing sentences and moved the introduction of the LAQ threshold to the Appendix, where we could give a more detailed explanation and comparison. Furthermore, we have focused more on our **novelty**: **we derive a new optimal quantization level selection by minimizing the upper bound of this model deviation caused by update skipping, and we propose a new skip rule which guarantees a better convergence rate than LAG.**
- **Methodology**: We have divided the previous Section 3 into two subsections, covering the optimal selection of the quantization level and the precise lazy aggregation criterion separately. In Section 3.1, as *Reviewer qabV* mentioned, we have specifically exhibited how to derive the optimal quantization level. And in Section 3.2, we have corrected a symbol error ($\boldsymbol{\theta}^{k+1}$) and updated our skip rule. Moreover, we have also moved Algorithm 1 to the main part of the paper to help readers better understand our proposed procedure.
- **Theoretical analysis**: On the basis of the original theoretical analysis, we have newly added the convergence rate comparison with LAG, which proves that our convergence rate is better.
- **Evaluation**: Inspired by *Reviewer NDFY*, we have enlarged the scale of the FL settings. Specifically, we have conducted the evaluation of **100** devices for the CIFAR-10 and CIFAR-100 datasets and **80** devices for the WikiText-2 dataset on 4 NVIDIA RTX A6000 GPUs (total 192G VRAM). Moreover, we have added a new Section 5.4 to interpret the effect of the tuning factor $\beta$ as an ablation study. And we have also implemented a fixed-bit quantization method (QSGD) for all FL settings, demonstrating that AQUILA still performs best among all the compared algorithms.

Moreover, since the skip rule is changed, we have updated our python **code implementation** in the supplementary material to facilitate reproducibility.

We hope that we have cleared all your concerns, and we will be happy to provide further information if needed.

---

> ### Author Response · Authors · 2022-11-18
> **Equation number changed**
>
> Some equation numbers changed during the revision. Here we provide a changed equation number list.
>
> Before revision → After revision
>
> (5) → (24)
>
> (8) → (6)
>
> (9) → (32)
>
> (11) → (12)
>
> Assumption 4.3 → Assumption B.1

---

### Decision · Program_Chairs · 2023-01-20

**Decision:**

Reject

**Justification For Why Not Higher Score:**

If the theory covered the case that was actually run in experiments (with stochastic gradients), then the paper would be stronger, but this seems like a major challenge. Without this, the primary contribution is the experiments, and these don't sufficiently help justify or explain why the proposed approach performs better than previous approaches. The reviewers weren't comfortable recommending acceptance (and I agree) without this.

**Justification For Why Not Lower Score:**

N/A

**Metareview: Summary, Strengths And Weaknesses:**

This paper presents an approach, AQUILA, to reduce communication overhead in federated learning (FL) by combining adaptive quantization (adapting the bit-rate) and lazily-aggregated gradients. The paper provides a theoretical convergence guarantees and an experimental evaluation illustrating the potential communication reduction achieved with AQUILA.

**Strengths:** We acknowledge and appreciate that there is some algorithmic novelty in the work, although it involves combining many ideas that have previously appeared in the literature. We also appreciate the experiments that were done to compare to previous work, including experiments added during the rebuttal period

**Weaknesses:** There are two main weaknesses of this work:
1. Because the theory focuses on deterministic gradients (the extension to handle stochastic gradients is not at all obvious) and largely combines existing proof techniques, this creates a significant gap between theory and practice, and there wasn't enough theoretical support on its own to justify accepting the paper; this effectively raises the bar on expectations for the experimental evidence.
2. Although the experiments include comparisons with many other methods in the literature and show promising performance, they don't help illustrate or understand why AQUILA performs better than previous approaches. For example, in Fig 2, we see that transmitted bits / epoch is relatively flat throughout training (with few exceptions), leading to the question of whether a simpler method like LAQ could perform nearly as well if it was tuned better.

In addition, the paper could be strengthened by improving clarity of presentation. One concrete suggestion is to clarify the description of the algorithm based on the reviewers' feedback. Another suggestion is to directly acknowledge the limitation of focusing the analysis on the deterministic setting (for example, in Section 6) and point this out as an important direction for future work. (To be clear, we understand that analyzing AQUILA with stochastic gradients would require substantial additional work and did not expect that for this submission.)

**Summary Of Ac-Reviewer Meeting:**

We had a virtual meeting to discuss this paper. All reviewers attended and contributed significantly to the discussion.

The reviewers discussed the merits and strengths of the work, especially the promising empirical results.

They also discussed the main concerns that had been raised in their reviews, and the author responses. In particular, the author responses weren't sufficient to convincingly address the concerns.

One reviewer also expressed that, although they found the paper acceptable, their confidence was lower because of being less familiar with this literature. Other reviewers that were more familiar with the literature were less favorable about the novelty of the work.

Overall, my final decision is based on the two main points mentioned under weaknesses in the meta-review, which were discussed extensively with the reviewers. If the theory covered the case that was actually run in experiments (with stochastic gradients), then the paper would be stronger, but this seems like a major challenge. Without this, the primary contribution is the experiments, and these don't sufficiently help justify or explain why the proposed approach performs better than previous approaches. The reviewers weren't comfortable recommending acceptance (and I agree) without this.